# CEREBROVOICE: A STEREOTACTIC EEG DATASET AND BENCHMARK FOR BILINGUAL BRAIN-TO-SPEECH SYNTHESIS AND ACTIVITY DETECTION

## ABSTRACT

Brain signal to speech synthesis offers a new way of speech communication, enabling innovative services and applications. With high temporal and spatial resolution, invasive brain sensing such as stereotactic electroencephalography (sEEG) becomes one of the promising solutions to decode complex brain dynamics. However, such data are hard to come by. In this paper, we introduce a bilingual brain-to-speech synthesis (CerebroVoice) dataset: the first publicly accessible sEEG recordings curated for bilingual brain-to-speech synthesis. Specifically, the CerebroVoice dataset comprises sEEG signals recorded while the speakers are reading Mandarin Chinese words, English words, and Mandarin Chinese digits. We establish benchmarks for two tasks on the CerebroVoice dataset: speech synthesis and voice activity detection (VAD). For the speech synthesis task, the objective is to reconstruct the speech uttered by the participants based on their sEEG recordings. We propose a novel framework, Mixture of Bilingual Synergy Experts (MoBSE), which uses a language-aware dynamic organization of low-rank expert weights to enhance the efficiency of language-specific decoding tasks. The proposed MoBSE framework achieves significant performance improvements over current state-of-the-art methods, producing more natural and intelligible reconstructed speech. The VAD task aims to determine whether the speaker is actively speaking. In this benchmark, we adopt three established architectures and provide comprehensive evaluation metrics to assess their performance. Our findings indicate that low-frequency signals consistently outperform high-gamma activity across all metrics, suggesting that low-frequency filtering is more effective for VAD tasks. This finding provides valuable insights for advancing brain-computer interfaces in clinical applications. The CerebroVoice dataset and benchmarks are publicly available on Zenodo and GitHub for research purposes.

## 1 INTRODUCTION

Recent advancements in brain-computer interfaces (BCIs) have opened a new frontier in human-computer interaction: speech synthesis directly from neural signals (1; 2; 3). Such systems hold significant potential to provide a natural means of communication for individuals with speech loss (4). Although surface EEG is a widely used non-invasive technique, it primarily captures cortical activity and lacks the spatial resolution needed to probe deeper brain regions, which are essential for speech production (5; 6). Given the complex nature of speech, brain-to-speech synthesis relies on the high-resolution and high signal-to-noise ratio, intracranial electroencephalography (iEEG) to capture intricate neural correlates of speech production (6; 7; 8).

The iEEG signal, also referred to as electrocorticography (ECoG) when using subdural grid electrodes or stereotactic EEG (sEEG) when using depth electrodes, has attracted interests across diverse domains of human neuroscience (9; 10). Many efforts have been made to ECoG-based brain-to-speech synthesis and achieved promising outcomes (6; 7; 11). On the other hand, sEEG has several unique advantages for brain-to-speech synthesis. Firstly, the implantation of sEEG electrode shafts into the brain involves smaller incisions, potentially with fewer complications (12). This offers a safer alternative for long-term brain activity monitoring (13; 14). Additionally, sEEG electrodes are placed directly within the brain tissue, allowing for more precise localization of functional areas (15).

Therefore, sEEG typically provides higher spatial resolution than ECoG. Furthermore, while ECoG offers high-density coverage of specific regions, sEEG provides sparse sampling across multiple geometric regions. This characteristic presents significant potential for speech synthesis that involves processes in deep brain regions or spatially disparate, bilateral areas (16; 17; 18; 3). Recent progress has also validated the feasibility and effectiveness of sEEG-based speech synthesis (19; 20; 21)

However, a major challenge is that iEEG signals are typically only available in clinical settings, limiting data collection due to the clinical environment and the underlying pathological conditions of participants. Thus, publicly available datasets, especially sEEG-speech parallel data, are extremely rare. This motivates the need for high-quality sEEG-speech parallel datasets.

The contributions of this work are as follows. we introduce the CerebroVoice dataset, the first publicly accessible sEEG recordings curated for bilingual brain-to-speech synthesis. The dataset includes sEEG signals recorded while speakers read Mandarin Chinese words, English words, and Mandarin Chinese digits. We establish benchmarks for two tasks: speech synthesis and VAD. For speech synthesis, we propose the Mixture of Bilingual Synergy Experts (MoBSE) framework, which dynamically organizes low-rank expert weights for more effective language-specific decoding. MoBSE shows significant performance improvements over current state-of-the-art methods, producing more natural and intelligible speech. For VAD, we reproduce three classic EEG-based architectures and provide comprehensive evaluation metrics, finding that low-frequency signals outperform high-gamma activity, suggesting low-frequency filtering is better suited for VAD tasks.

## 2 RELATED WORK

Considerable progress has been made in iEEG (sEEG and ECoG) based brain-to-speech synthesis in recent years. Martin et al. (22) decoded spectro-temporal features of speech from brain activity using ECoG, and Mugler et al. (23) further demonstrated that the full set of American English phonemes can be decoded from ECoG. In (11), Moses et al. explored real-time decoding of perceived and produced speech from high-density ECoG activity during a question-and-answer dialogue task. Angrick et al. (24) explored the use of deep neural networks (3D convolutional neural networks) for reconstructing speech from ECoG recordings. Moses et al. (4) investigated the long-term stability of ECoG recording and its performance in decoding speech over an extensive 81-week recording period in a paralyzed patient with anarthria.

More recently, Metzger et al. (6) have further improved the performance of speech decoding using ECoG collected over 13 days. Building on this study, Feng et al. (25) further conducted similar work in Mandarin Chinese. Despite much progress, the datasets for these studies are not publicly available. The absence of publicly released datasets hinders reproducibility and collaborative research efforts in brain-to-speech synthesis.

Similarly, publicly available sEEG-speech datasets remain scarce, as summarized in Table 1. Angrick et al. (8) released a 15-minute sEEG-speech dataset from one single Dutch-speaking epilepsy patient, while Kohler et al. (26) published a similar dataset of three epilepsy patients, with 10 to 20 minutes each. Verwoert et al. (13) also released a dataset of 10 Dutch-speaking epilepsy patients, however, each only contributed 5 minutes of data. The above sEEG data are not adequate for machine learning studies. To address this, a recent dataset release offers 3 hours of sEEG-speech data per subject (27). However, most prior brain-to-speech synthesis research has focused on monolingual tasks, with little exploration of bilingual speakers. The development of an iEEG-based encoder for bilingual speech synthesis is highly desirable (28). This gap underscores the necessity of an sEEG dataset specifically designed for bilingual speech synthesis.

Additionally, there is limited research on VAD using sEEG, with no publicly available datasets specifically tailored for this task (29; 30). Consequently, we established a benchmark and compared three classical baseline models to evaluate their performance.

Addressing the research need, we propose a CerebroVoice dataset, comprising sEEG recordings captured when the participant read aloud Mandarin Chinese words, English words, and Mandarin Chinese digits. Two patients, both implanted with depth electrodes to identify epileptic foci and plan potential resections, were recruited for this study. As shown in Table 1, each participant's data recording duration was about 75 minutes. This CerebroVoice dataset represents the first bilingual iEEG-speech dataset encompassing both tonal (Mandarin Chinese) and non-tonal (English) languages.

This unique feature significantly contributes to advancing research in the field of brain-to-speech synthesis.

Table 1: A summary of our proposed CerebroVoice and other existing publicly available sEEG-based brain-to-speech synthesis datasets.

| Year | 2021 | 2021 | 2022 | 2024 | 2024 |
|---|---|---|---|---|---|
| Publication | Communications Biology (8) | Neurons Behavior (26) | Scientific Data (13) | NeurIPS (27) | CerebroVoice (this work) |
| Participants | 1 | 3 | 10 | 12 | 3⋆ |
| No. of Electrodes | 128 | 117-127 | 56-234 | 72-158 | 176-185 |
| Language | Dutch | Dutch | Dutch | Chinese | Chinese & English |
| Speaking | Words | Sentences | Words | Words | Words & digits |
| Duration per Person | 15 | 10-20 mins | 5 mins | 180 mins | 75 mins |
| Task | Speech Synthesis | Speech Synthesis | Speech Synthesis | Word Classification | Speech Synthesis, VAD |

⋆ More data are continuously added to the CerebroVoice dataset as new patients join the study.

# 3 CEREBROVOICE DATASET CONSTRUCTION

## 3.1 PARTICIPANTS

Two patients with epilepsy undergoing neurosurgical treatment were enrolled as the listening and speaking subjects in the data collection. They are referred to as the participants. One participant (Subject 1) was a 25-year-old male native Mandarin Chinese speaker with basic English conversation skills. The other participant (Subject 2) was a 30-year-old female native Mandarin Chinese speaker with limited English proficiency.

The study was conducted in accordance with the principles embodied in the Declaration of Helsinki and approved by the Ethics Committee of the South China Hospital of Shenzhen University (HNLS20231229003-A). Both patients gave written informed consent to participate in the study. Data collection was conducted under the supervision of experienced doctors to ensure the comfort and safety of the participants. During the recording process, patients were required not to enter any personal identification information. Therefore, this dataset does not contain the identity information of actual users.

## 3.2 NEURAL RECORDINGS

Both participants were implanted with sEEG electrode shafts to identify epileptogenic foci and all the locations of sEEG electrodes were determined based on each patient's specific epilepsy treatment plan. 13 electrode shafts were implanted in each subject. Each shaft contains 8-16 electrode contacts, resulting in a total of 176 and 185 electrode contacts for Subjects 1 and 2, respectively. To accurately determine the positions of contacts, we used an open-source MATLAB package LeGUI (31), in which the processing is performed based on Statistical Parametric Mapping toolbox (SPM12) (32). Fig in appendix illustrates three views of the depth electrode locations for each participant, where dots of the same color represent electrodes belonging to the same shaft. Notably, all electrodes in Subject 1 were implanted within the right hemisphere, while those in Subject 2 were located in the left hemisphere.

## 3.3 DATA ACQUISITION

The participants underwent implantation of platinum-iridium sEEG electrode shafts (Sinovation (Beijing) Medical Technology SDE-10/12/16, China), featuring a diameter of 0.8 mm and an inter-contact distance of 3.5 mm. Each electrode shaft contained between 10 and 16 electrode contacts. Notably, the placement of all electrodes was determined based on the patients' therapeutic requirements. sEEG signals were recorded at a sampling rate of 1000 Hz or 500 Hz (Nihon Kohden EEG 1200, Tokyo, Japan), and auditory data was simultaneously collected. Specifically, audio recordings were captured with a JABRA speakerphone using OBS Studio software at 48 kHz.

As depicted in Fig. 1, a computer was placed in front of the participants, serving as the control center. It delivered the audio stimuli via a speaker, and recorded the participant's speech. During recording, the computer screen shows a blank screen so as not to distract the participants. Both the participants' sEEG signals and audio signals were recorded. To ensure synchronization between the auditory stimuli and sEEG responses, we employed a Python-scripted tool to play audio stimuli and simultaneously mark the corresponding sEEG responses.

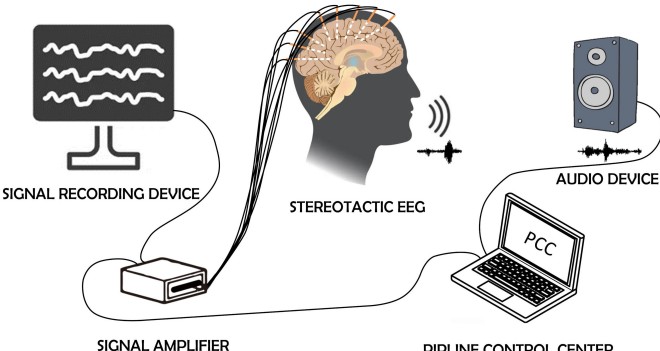

Figure 1: Experiment setup for CerebroVoice data collection. sEEG and speech are recorded simultaneously while a participant speaks Mandarin Chinese words, English words, and Mandarin Chinese digits.

## 3.4 EXPERIMENT PROTOCOL

During our experiment, participants were presented with auditory stimuli across three different categories: 30 categories of Mandarin Chinese words, 10 categories of Mandarin Chinese digits (1-10), and 10 categories of English words. The duration designated for listening and repeating was set at 5 seconds for both Mandarin Chinese and English words, while for Mandarin Chinese digits, it was set to 4 seconds. Each participant completed 8 rounds of experiments, with each round consisting of 30 English words, 60 Mandarin Chinese digits, and 110 Mandarin Chinese words. At the beginning of each round, a participant is given a 5-second interval to get ready, where a prompt "Please listen to the audio attentively and repeat loudly what you will hear" is played, that is followed by a "ding" sound to signal the start of the attended speech content. After each word was played, the participants were expected to recite the speech content within 1.5 seconds, and then remain relaxed until the next "ding" sounded.

To avoid fatigue, the participants took a 5 to 10-minute break between two rounds. Additionally, several familiarization trials were conducted to ensure that the subjects understood the experimental procedures before recording. Following data collection, we assessed the quality of the audio recordings, manually removing recordings with mispronunciations or pauses. First, we employed a pre-trained Automatic Speech Recognition (ASR) model to transcribe the speech into text. We then compared this transcription with the ground truth text to calculate the Word Error Rate (WER). For samples where the WER was not 100%, a manual review was conducted to determine whether discrepancies were due to reading errors or ASR system inaccuracies. As a result, the CerebroVoice dataset comprises 72.94 minutes of data for Subject 1 and 76.49 minutes for Subject 2.

## 4 DATA PREPROCESSING

### 4.1 DATA LOADING

The CerebroVoice dataset is publicly available for research use (https://zenodo.org/records/13332808). To simplify the use of the data, we have preprocessed the sEEG signals and corresponding speech signals. Specifically, files with the extension _SEEG.npy contain the processed sEEG data for each participant, while files ending in _MEL.npy contain the corresponding mel-spectrogram of the speech.

### 4.2 NEURAL SIGNAL PREPROCESSING

First, we excluded electrodes identified in epileptologists' reports as showing abnormal epileptiform discharges (33). Specifically, 62 electrodes were removed from Subject 1 (114 left) and 27 electrodes from Subject 2 (158 left). Subsequently, bipolar referencing was applied to the remaining sEEG signals (27). Previous studies have highlighted the critical role of high-gamma frequency (HGA) and low-frequency signal (LFS) features in synthesizing speech from brain signals (8; 34; 6). Accordingly, we followed the preprocessing methods used in previous research to extract the LFS and HGA

frequency bands (6). Additionally, we tested broadband signals (BBS), which combine both LFS and HGA sEEG features, to provide a comprehensive perspective and evaluate their combined contributions to speech synthesis performance. Specifically, to compute HGA, we first band-passed the signals in the high-gamma frequency range (70–150 Hz), then calculated the analytic amplitude of these signals, and finally downsampled them to 200 Hz. For LFS, we applied a low-pass anti-aliasing filter with a cutoff frequency of 100 Hz before downsampling the signals to 200 Hz. Lastly, we normalized the extracted HGA and LFS signals from each sEEG electrode within each 1.5-second window.

### 4.3 Audio Signal Preprocessing

We used LibROSA, a commonly adopted Python library for audio processing (35), to downsample the audio signals to 16 kHz and extract the mel-spectrograms. To capture the temporal dynamics of the audio signal, a window length of 64 milliseconds and a hop length of 20 milliseconds were set. Additionally, we set the number of bins in the mel-spectrogram to 80, aiming to capture sufficiently detailed frequency information to describe the participants' speech signals (36).

### 4.4 Data Preparation for Voice Activity Detection

We implemented VAD using the Mel-Filter Bank and Energy-based VAD methods. The Mel-Filter Bank transforms the audio signal into mel-scaled spectrograms, while the Energy-based VAD processes the log-energy of Mel-Frequency Cepstral Coefficients (MFCCs) to detect speech activity. Key parameters include a window length of 0.064 seconds and a window shift of 0.02 seconds, which define the audio segmentation into overlapping frames.

## 5 Experiment

### 5.1 Baseline Methods for Speech Synthesis

#### 5.1.1 Baseline Architectures

sEEG-based brain-to-speech study is still at its early stage. We propose an sEEG to mel-spectrogram conversion model based on FastSpeech2, which is a state-of-the-art text-to-speech synthesis framework with an encoder-decoder structure. The model architecture is shown in Fig. 2.

In the original FastSpeech2, text embeddings are used as input to the encoder. For our sEEG-based speech synthesis task, we replaced these text embeddings with embeddings derived from sEEG signals. Specifically, we transformed 1.5-second sEEG signals into a 2D data format with dimensions (75, C), where 75 represents the time dimension and C represents the channel dimension. This transformation is analogous to the mel-spectrogram features, which have dimensions of (75, 80) in which 80 is the dimension of features and each frame lasts 0.02-second, ensuring alignment with the temporal structure of the speech.

The FastSpeech2-based model first maps high-dimensional sEEG signals to a lower-dimensional space through an embedding layer. Subsequently, these embedded signals are further processed by positional encoding to obtain the positional information of the time series. The encoder extracts deep features, and the decoder decodes based on these features to ultimately output the mel-spectrogram. We will elaborate the process in detail next.

#### 5.1.2 Encoder

The encoder is implemented in a Transformer architecture which follows that in the FastSpeech2 model, utilizing six feedforward Transformer (FFT) blocks (37). These FFT blocks, through the self-attention mechanism and position-wise feedforward networks, enhance the model's ability to capture long-distance dependencies. Each FFT block contains a self-attention layer and a feedforward network layer that can effectively encode the temporal characteristics of the sEEG signal.

Figure 2: Overview of the Bilingual sEEG-based Speech Decoding Framework. (a) The pipeline for generating speech from sEEG signals (b) The module unit of the encoder used in FastSpeech2. (c) The approach used by FastSpeech2 for simultaneous bilingual decoding (d) The proposed MoBSE (Mixture of Bilingual Synergy Experts) structure, which employs multiple low-rank experts with dynamically organized expert weights informed by language-aware priors.

### 5.1.3 MEL-SPECTROGRAM DECODER

The decoder is implemented with a single one-dimensional convolutional layer to directly transform the encoded high dimensional features into mel-spectrogram features. The final output dimension of the mel-spectrogram features is (75,80), where a speech segment of 1.5 seconds consists of 75 speech frames of 20 milliseconds each, and there are 80 elements in a mel-spectrogram feature frame. These features together constitute the spectral representation of the audio signal.

### 5.1.4 WAVEFORM DECODER

Since the advent of WaveNet (38) in 2016, neural vocoders have played a crucial role in reconstructing highly natural speech, capable of converting a mel-spectrogram frame into high quality speech waveform. In this study, we used the HiFi-GAN vocoder (39), which consists of a generator and two discriminators: multi-scale and multi-period discriminators. This vocoder is pretrained in advance.

### 5.1.5 TRAINING DETAILS

We adopt the positional encoding scheme as in FastSpeech2. The introduction of positional encoding enables the model to more effectively capture and understand the specificity of different channels in the sEEG signal, as well as the temporal information at different moments of the sequence. It is expected that this encoding helps distinguish the unique physiological signals carried by each channel, at the same time, identifies the characteristics of the signal as it changes over time, which is crucial for accurately parsing the temporal structure of sEEG signals.

In the training process, we adapted the training methodology to fit our task requirements. The Adam optimizer was utilized with hyperparameters $\beta_1 = 0.9$ and $\beta_2 = 0.98$. The model was trained using a batch size of 16 and a learning rate of 0.001. The L1 loss function was adopted to measure the difference between the predicted and ground-truth mel-spectrograms.

## 5.2 MIXTURE OF BILINGUAL SYNERGY EXPERTS

As illustrated in Figure. 2 (d), this module is designed for the mixture of bilingual synergy experts within the feed-forward network (FFN) of FasterSpeech2. It is specifically tailored for the task of bilingual stereo-electroencephalography (sEEG)-based speech decoding, enhancing the model's ability to process and decode bilingual information from sEEG signals.

The input to this module is a feature tensor $\mathbf{x} \in \mathbb{R}^{B \times T \times D}$, where $B$ represents the batch size, $T$ denotes the temporal dimension, and $D$ is the feature dimension. The features encapsulate temporal sEEG information from two languages. To effectively decode the bilingual information, we employ a mixture of experts framework, where each expert is specialized in extracting features specific to one language's sEEG signals.

For each task label, indicating whether the decoding task is for Mandarin or English and represented as a one-hot encoded vector $\mathbf{t}$, we perform a linear transformation. The transformed vector is fused with the input features $\mathbf{x}$ and passed through a linear layer, followed by global average pooling (GAP) over the temporal dimension to obtain the input $\mathbf{g}$ for the gating network: $\mathbf{g} = \text{GAP}(\text{Linear}(\mathbf{x} + \text{Linear}(\mathbf{t})))$, where $\mathbf{x}$ denotes the feature tensor, $\mathbf{t}$ represents the one-hot encoded task label, $\text{Linear}$ signifies a linear layer, and GAP signifies global average pooling over the temporal dimension.

The gating network, parameterized by a MLP, converts the bilingual fused features into weights $w$ for each expert: $w = \text{GatingNetwork}(\mathbf{g})$. The final output $\mathbf{y}$ is then obtained by combining the weights with the outputs from each low-rank expert: $\mathbf{y} = \sum_{i=1}^{N} w_i \cdot \text{LowRankExpert}_i(\mathbf{x})$, where $i$ represents the $i$-th expert in the mixture of experts framework. $\text{LowRankExpert}$ comprises a dimension reduction and an expansion linear layer.

We chose to use 8 experts in the MoBSE framework based on results from ablation studies, which tested configurations with 4, 6, 8, 10, and 12 experts, the configuration with 8 experts achieved the best overall performance, striking a balance between effective language-specific decoding and minimizing redundancy or overfitting.

This architecture ensures that the respective language experts can process the corresponding sEEG information with high effectiveness. The mechanism guarantees accurate decoding of bilingual sEEG features, leveraging the unique strengths of language-specific experts. This innovative approach significantly enhances the model's adaptability and performance in bilingual speech decoding tasks, positioning it as a robust solution for future research and application in the field of neural decoding.

### 5.3 SPEECH SYNTHESIS METHODS FOR COMPARISON

We evaluate the performance of various speech synthesis models using the CerebroVoice dataset. We employ metrics such as Pearson Correlation Coefficient (PCC) (40), Mel Cepstral Distortion (MCD) (40), Root Mean Square Error (RMSE) (40), and Short-Time Objective Intelligibility (STOI) (41) to assess the effectiveness of each model. Specifically, we compare BrainTalker (40), FastSpeech 2 (42), Shaft CNN (19), Hybrid CNN-LSTM (43), Dynamic GCN-LSTM (44), and our proposed Mixture of Bilingual Synergy Experts (MoBSE) framework.

### 5.4 VOICE ACTIVITY DETECTION METHODS FOR COMPARISON

We utilized three classical baseline methods for VAD: EEGNet (45), STANet (46), and EEG-ChannelNet (47). EEGNet is designed for EEG data, using depthwise separable convolutions to capture spatial features efficiently. STANet (Spatial-Temporal Attention Network) employs attention mechanisms to model spatial and temporal dependencies, improving detection robustness. EEG-ChannelNet uses channel attention to selectively aggregate information from different EEG channels.

## 6 RESULTS AND DISCUSSION

### 6.1 EVALUATION OF SYNTHESIZED SPEECH

To evaluate the performance of sEEG-based speech synthesis in the CerebroVoice dataset, we compare the mel-spectrograms and waveforms of the reconstructed and original spoken speech samples. Like in previous studies (13; 23), we use the Pearson Correlation Coefficient (PCC) to assess the similarity between the reconstructed and original mel-spectrograms.

As illustrated in Figs. 3 (a) and (b), the synthesized speech samples closely resemble the spoken speech samples, with some detail lost in the mel-spectrogram representations. Table 2 further summarizes the performance of different sEEG features (BBS, HGA, and LFS) for predicting speech using FastSpeech2 or MoBSE, respectively. Statistically significant improvements with our proposed MoBSE over current state-of-the-art methods were observed across all BBS, HGA, and LFS signals (paired $t$-test, $p < 0.05$).

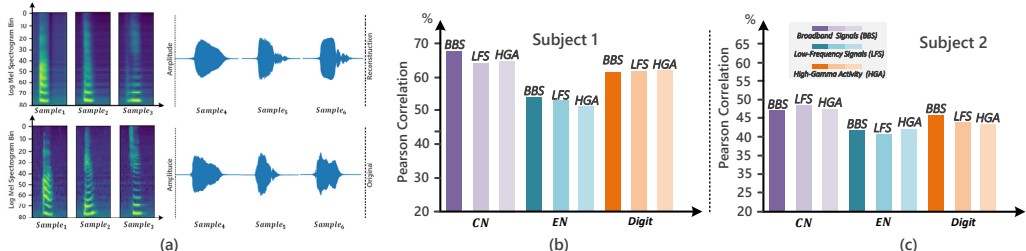

(a)                                            (b)                                            (c)

Figure 3: Speech decoding performance of the proposed CerebroVoice. (a) Comparison of mel-spectrograms and waveforms for 6 words from Subject 2. (b-c) Pearson Correlation Coefficient for reconstructed vs. original mel-spectrograms across Mandarin Chinese (CN) and English (EN) words using different sEEG features.

Table 2: Comparative analysis of the speech synthesis performance of different spoken word categories and sEEG features for Subjects 1 and 2. The Pearson Correlation Coefficients (PCC) between the reconstructed and original mel-spectrograms are reported with better results between FastSpeech2 and MoBSE in bold font.

| sEEG feature | Methods | Subject 1 | | | | Subject 2 | | | |
|---|---|---|---|---|---|---|---|---|---|
| | | Chinese | English | Digit | Avg | Chinese | English | Digit | Avg |
| LFS | FastSpeech2 | **0.647** | 0.492 | 0.585 | 0.574 | **0.483** | 0.328 | 0.437 | 0.416 |
| | MoBSE(Ours) | 0.638 | **0.531** | **0.615** | 0.575 | 0.473 | **0.406** | **0.448** | 0.442 |
| HGA | FastSpeech2 | **0.658** | 0.450 | **0.618** | 0.575 | **0.474** | 0.381 | **0.433** | 0.429 |
| | MoBSE(Ours) | 0.642 | **0.513** | 0.599 | 0.585 | 0.460 | **0.422** | 0.431 | 0.438 |
| BBS | FastSpeech2 | 0.655 | 0.469 | **0.612** | 0.578 | **0.472** | 0.390 | 0.452 | 0.438 |
| | MoBSE(Ours) | **0.673** | **0.537** | 0.602 | 0.604 | 0.455 | **0.441** | **0.459** | 0.452 |

### 6.1.1    COMPARISON OF DIFFERENT SUBJECTS

The decoding performance of Subject 1 surpasses that of Subject 2 across all spoken word categories (Mandarin Chinese, English, digits). As shown in Table 4, the average PCC correlations of Subject 1 using different sEEG features are 0.598 for LFS, 0.596 for HGA, and 0.607 for BBS, respectively, while those of Subject 2 are 0.446, 0.431, and 0.457, respectively. This can be explained by the variability in sEEG signals among different subjects, influenced by factors such as individual differences, signal quality, electrode placement, and participant concentration (13; 26; 12).

### 6.1.2    COMPARISON OF SPOKEN WORD CATEGORIES

The performance of speech synthesis also varies across different spoken word categories. It can be observed that Subject 1 performs the best in the speech decoding of Mandarin Chinese words, with an average PCC of 0.652, while the average PCC for English words and Mandarin Chinese digits are 0.499 and 0.605, respectively. Similarly, Subject 2 exhibits consistent decoding performances across all three spoken word categories, with higher PCC in decoding Mandarin Chinese words compared to English words and Mandarin Chinese digits.

These results suggest that decoding Mandarin Chinese words from sEEG signals might be easier for our CerebroVoice dataset, possibly due to both participants being native Mandarin Chinese speakers. Additionally, the larger training sample sizes of Mandarin Chinese words could be another reason. Notably, the number of Mandarin Chinese is more than twice that of English words and Mandarin Chinese digits.

### 6.1.3    COMPARISON OF DIFFERENT SEEG FEATURES

Additionally, we investigate the performance of different sEEG features (BBS, HGA, and LFS) for predicting speech, as shown in Fig. 3. It can be observed that the BBS feature exhibits superior performance, with an average PCC of 0.518 across all spoken word categories, followed by HGA

and LFS features. One possible explanation could be that BBS feature integrates both high and low-frequency information of sEEG, thus enabling a more comprehensive representation of speech features. Moreover, HGA feature outperforms the LFS for both subjects. These findings align with previous research, suggesting that high gamma band brain activity contains highly localized information relevant to speech (6; 48; 49) and language (50) processes.

Table 3: Comparison of MoBSE with other state-of-the-art methods across different subjects.

| Subjects | Model | PCC↑ | STOI↑ | MCD↓ | RMSE↓ |
|----------|-------|------|-------|------|-------|
| Subject 1 | Brain Talker | 0.584 | 0.193 | 4.282 | 0.523 |
| | MoBSE(Ours) | **0.604** | **0.285** | **4.143** | **0.501** |
| | Shaft CNN | 0.583 | 0.195 | 4.358 | 0.548 |
| | Hybrid CNN-LSTM | 0.564 | 0.170 | 4.448 | 0.562 |
| | Dynamic GCN-LSTM | 0.551 | 0.153 | 4.556 | 0.583 |
| Subject 2 | Brain Talker | 0.434 | 0.142 | 5.958 | 0.635 |
| | MoBSE(Ours) | **0.452** | **0.184** | **5.652** | **0.622** |
| | Shaft CNN | 0.432 | 0.153 | 5.986 | 0.644 |
| | Hybrid CNN-LSTM | 0.424 | 0.126 | 6.124 | 0.656 |
| | Dynamic GCN-LSTM | 0.408 | 0.122 | 6.334 | 0.660 |

### 6.1.4 COMPARING VARIOUS STATE-OF-THE-ART METHODS ON OUR PROPOSED CEREBROVOICE DATASET

we conducted an ablation analysis to compare the performance of MoBSE with other state-of-the-art methods, including Brain Talker, Shaft CNN, Hybrid CNN-LSTM, and Dynamic GCN-LSTM, across two subjects. Our analysis focused on key performance metrics: Pearson Correlation Coefficient (PCC), Short-Time Objective Intelligibility (STOI), Mel Cepstral Distortion (MCD), and Root Mean Square Error (RMSE).

For Subject 1, MoBSE outperformed other models with the highest PCC of 0.604 and STOI of 0.285, indicating improved correlation and intelligibility of the reconstructed speech. Additionally, MoBSE achieved the lowest MCD of 4.143 and RMSE of 0.501, demonstrating superior accuracy and reduced distortion in speech reconstruction. Similarly, for Subject 2, MoBSE maintained its leading performance with a PCC of 0.452 and a STOI of 0.184, along with the lowest MCD of 5.652 and RMSE of 0.622. These results consistently show that MoBSE provides a significant improvement in speech quality and intelligibility compared to other methods.

### 6.1.5 COMPARING SPEECH DEMOS DECODED FROM CEREBROVOICE WITH THOSE FROM OTHER PAPERS

We conducted an ablation analysis to compare the quality of speech generated by our CerebroVoice system with outputs from existing research, specifically NMI-24 (51) and SD-22 (13). In a subjective Mean Opinion Score (MOS) test, using a 1-5 scale, 15 raters evaluated the speech samples based on a combination of naturalness and intelligibility. CerebroVoice achieved an average score of 4.33, demonstrating superior performance compared to NMI-24, which scored 2.93, and SD-22, which scored 1.27. These results indicate that CerebroVoice generates speech perceived as both more natural and intelligible.

For the objective evaluation, we utilized the NISQA metric, a no-reference speech quality assessment tool. CerebroVoice obtained a score of 3.2751, while NMI-24 and SD-22 scored 2.2828 and 1.8911, respectively. The alignment between subjective and objective evaluations highlights the superior quality of speech produced by CerebroVoice compared to existing research. This analysis underscores the advancements in speech quality achieved by our system.

## 6.2 EVALUATION & HIGHLIGHT OF VOICE ACTIVITY DETECTION

The VAD accuracy is evaluated by computing the ratio of the number of correctly predicted windows to the total number of windows. In this measurement, the window length is set to be 0.064 seconds.

Table 4: Comparative analysis of the VAD performance using different sEEG features and baseline architectures for Subjects 1 and 2. The metrics reported are Balanced Accuracy and AUROC

| sEEG feature | Metrics | Subject 1 | | | Subject 2 | | |
|---|---|---|---|---|---|---|---|
| | | EEGNet | STANet | EEGChannelNet | EEGNet | STANet | EEGChannelNet |
| LFS | Balanced Accuracy | 0.792 | 0.782 | 0.811 | 0.660 | 0.651 | 0.684 |
| | AUROC | 0.852 | 0.856 | 0.905 | 0.712 | 0.699 | 0.752 |
| HGA | Balanced Accuracy | 0.660 | 0.624 | 0.755 | 0.589 | 0.587 | 0.626 |
| | AUROC | 0.722 | 0.684 | 0.834 | 0.620 | 0.622 | 0.675 |
| BBS | Balanced Accuracy | 0.807 | 0.735 | 0.850 | 0.672 | 0.646 | 0.730 |
| | AUROC | 0.867 | 0.806 | 0.928 | 0.724 | 0.695 | 0.803 |

LFS: Low-frequency signals (below 100 Hz)
HGA: High-gamma activity (between 70 and 150 Hz)
BBS: Broadband signals (combining both LFS and HGA sEEG features)

**Superior Performance of EEGChannelNet with BBS Features:** The EEGChannelNet architecture consistently demonstrated superior performance across both subjects and all sEEG features. Notably, it achieved the highest Balanced Accuracy (0.850 for Subject 1 and 0.730 for Subject 2) and AUROC (0.928 for Subject 1 and 0.803 for Subject 2) when using the Broadband Signals (BBS) feature. This indicates that combining both low and high-frequency sEEG features provides a more comprehensive representation of speech activity, enhancing the model's performance.

**Impact of Low-Frequency Signals (LFS):** Low-frequency signals (LFS) showed substantial effectiveness, particularly with EEGChannelNet, achieving a Balanced Accuracy of 0.811 and an AUROC of 0.905 for Subject 1. This suggests that low-frequency components of sEEG signals are crucial for accurately detecting voice activity, corroborating the findings that LFS outperforms high-gamma activity in VAD tasks.

**Variability Among Subjects:** The results highlight a significant variability in VAD performance between subjects. Subject 1 consistently outperformed Subject 2 across all metrics and sEEG features. For instance, the highest Balanced Accuracy for Subject 2 was 0.730 (BBS with EEGChannelNet), compared to 0.850 for Subject 1. This discrepancy underscores the importance of personalized calibration in brain-computer interface applications.

**Broadband Signals (BBS) as the Optimal Feature:** BBS features, which integrate both low and high-frequency information, emerged as the optimal feature set for VAD tasks. The average performance metrics for BBS were higher than those for LFS and HGA, indicating that a comprehensive approach to sEEG signal processing can significantly enhance VAD accuracy.

## 7 LIMITATION AND FUTURE WORK

In this study, the placements of sEEG electrodes were determined solely based on the patient's clinical needs. Hence, there was significant inter-individual variability in terms of brain regions. This variability is undesirable because it makes it difficult to compare results across participating subjects, and generalize to new subjects. To establish the broader applicability of the findings, we are looking into scaling up the data collection effort towards a larger cohort of participating subjects.

## 8 CONCLUSION

In this work, we introduced CerebroVoice, the first publicly accessible sEEG dataset for bilingual brain-to-speech synthesis and VAD. Contributed by two bilingual participants, this dataset supports the study of how spoken languages, word categories, frequency bands, and decoding models impact decoding accuracy. We validated the dataset's quality through benchmarks for speech synthesis and VAD tasks. Our proposed Mixture of Bilingual Synergy Experts (MoBSE) model significantly outperformed the FastSpeech2 baseline in speech synthesis, producing more natural and intelligible speech. For VAD, low-frequency signals proved superior to high-gamma activity, providing valuable insights for brain-computer interface applications. This study offers essential data and theoretical support for the research community, promoting interdisciplinary integration between neuroscience and artificial intelligence. The success of sEEG-based brain-to-speech synthesis and VAD tasks not only enhances our understanding of the human brain but also supports the development of innovative communication and diagnostic technologies.

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

# CerebroVoice: A Stereotactic EEG Dataset and Benchmark for Bilingual Brain-to-Speech Synthesis and Activity Detection Supplementary Material

This supplement to our main paper, "CerebroVoice: A Stereotactic EEG Dataset and Benchmark for Bilingual Brain-to-Speech Synthesis and Activity Detection," provides an in-depth explanation of the dataset collection methods and includes a comprehensive data card. It also outlines the licensing information for the dataset and includes an author statement verifying compliance with these licensing terms. Furthermore, it addresses the societal implications, providing a Preliminary Assessment and Disposal Plan of Relevant Risks as well as discussing Ethical Issues and Countermeasures. Detailed descriptions of the methods implemented on the dataset, along with the datasheets, are also included.

## 1    Data Collection

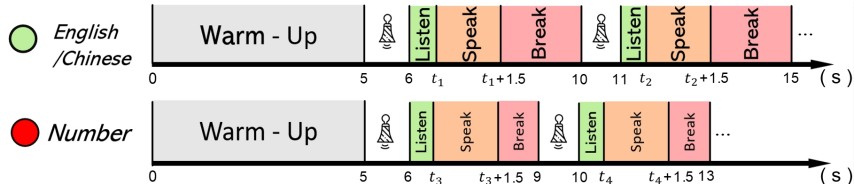

Figure 1: The timeline of experiment of each round

In our study, subjects were exposed to auditory stimuli from three different classifications: 30 categoriess of Chinese Mandarin words, 10 categoriess of Chinese Mandarin digits, and 10 categories of English words. The listening and repetition phase for both Chinese Mandarin and English words was allocated 5 seconds, whereas for Chinese Mandarin digits, this phase lasted 4 seconds. Participants underwent 8 rounds of the experiment, each round comprising 30 English words, 60 Chinese Mandarin digits, and 110 Chinese Mandarin words. At the start of each round, subjects had a 5-second preparation period, during which they were instructed through an audio prompt, "Please listen to the audio attentively and repeat loudly what you will hear," followed by a "ding" sound indicating the commencement of the speech content to be attended to. Following the playback of each word, subjects were required to repeat the speech content within 1.5 seconds and then stay relaxed until the next "ding" was heard. The data collection timeline for each round is depicted in Figure. 1.

### 1.1    Preliminary Assessment and Disposal Plan of Relevant Risks

To ensure the scientific property of the trial and the safety of the participants, we conducted a comprehensive assessment of the trial participants. Eligible trial participants were required to sign an informed consent form to understand the purpose, process, possible adverse reactions of the trial in detail, and clarify the relevant safety measures.

During the experiment, doctors and research teams worked together to ensure the safety and comfort of patients. If the patient felt tired during the trial, we would suspend the trial at any time to provide

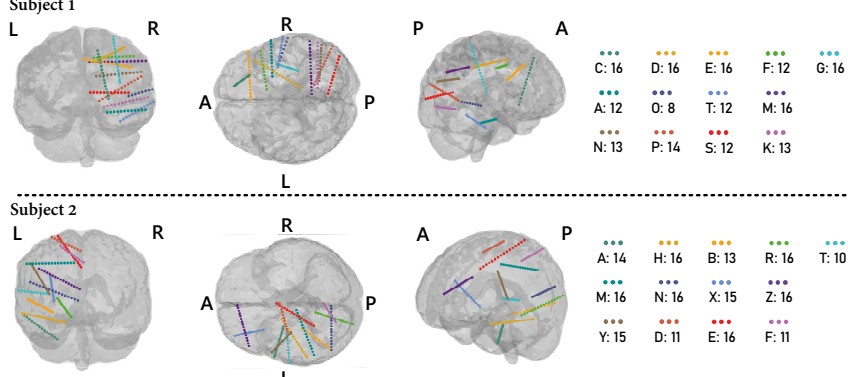

Figure 2: sEEG electrode contact locations for each subject. Dots of the same color represent electrode contacts positioned on the same electrode shafts. These locations are determined by co-registering pre-implantation magnetic resonance imaging (MRI) scans with post-implantation computed tomography (CT) scans.

rest. In addition, we closely monitored any potential risks during the trial and be ready to respond to emergencies at any time to maximize the safety and legal rights of the subjects.

## 1.2 Ethical Issues and Countermeasures

(1) Individuals participated in the study on a voluntary basis, and after ensuring that the subjects understand the relevant information, written informed consent were obtained from the subjects.

(2) All measures have been taken to protect the privacy of the subjects and keep personal information confidential.

(3) Each subject received sufficient information, including the purpose and methods of the study, any possible conflicts of interest, the researcher's organizational affiliation and potential risks, any discomfort that the study may cause, and any other information related to the study.

(4) Each subject was informed of his or her right to refuse to participate in the study and the right to withdraw consent to withdraw from the study at any time.

## 2 Dataset Structure

Our dataset collected 3200 samples from 3 volunteers, and then reserved 3069 samples, including 1493 samples from the first participant and 1576 samples from the second participant. Our data includes 27 folders. The outermost three folders are classified into BBS, HGA, and LFS to represent different frequency bands. The middle three folders are classified into Chinese Mandarin, English, and digits according to the type of words. It is essential to note that within each frequency band, we extracted samples from the initial pool of 3069, giving us a total of 9207 distinct samples across the full spectrum of frequency bands. This additional extraction process has allowed us to delve deeper into the data and create a comprehensive and detailed dataset.

As illustrated in Figure. 3, the innermost three folders are training set, validation set, and test set. In order to facilitate data users to view the basic information of each sample, we use a unified format to name the files of the training set, validation set, and test set, namely roundID_wordID_wordName, where round ID represents the round of experiments, word id represents the number of words read by the participant in this round of experiments, and word name represents the content of the words read by the participant. For ease of use, we provide the preprocessed sEEG signal and mel-spectrogram, both stored in npy format. It contains the following data:

(1) sEEG: a data matrix representing sEEG signals, ending with SEEG.npy, in the shape of T * F, where T represents the time dimension and F is the number of features. For HGA and LFS, the number of features is the same as the number of sEEG channels, and for BBS, the number of features is twice the number of channels. The number of valid channels for the first participant is 114, and the number of valid channels for the second participant is 158.

(2) Mel-Spectrogram: a data matrix representing the mel-spectroogram of audio signals, ending with MEL.npy, in the shape T*80, where T represents the time dimension and 80 represents the number of bin of the mel-spectrogram.

Additional dataset statistics are listed in Table 1. Note that the Total Number of Samples refers to the combined samples across all frequency bands (BBS, HGA, and LFS), while the Total Number of Words indicates the number of samples within any single frequency band.

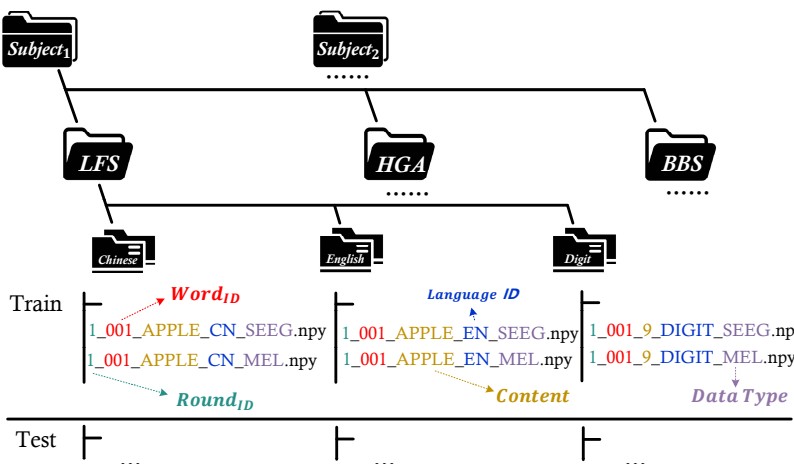

Figure 3: Dataset structure showing the organization of sEEG and audio data, in npy format.

| Category | Data |
|---|---|
| Total Number of Participants | 3 |
| Gender Ratio | 1:2 |
| Total Number of Sample | 9,207 |
| Total Number of Words | 3,069 |
| Number of Language | 2 |
| Number of Word Types | 3 |
| Number of Categories | 50 |

Table 1: CerebroVoice Dataset Card- This table enumerates dataset statistics, such as the total number of participants, gender ratio, total number of samples, total number of words, number of languages, word types, and categories. These factors collectively give an overview of the compiled dataset.

# 3 Societal Impact

As we point out in Section 7 of the paper, we publish a sEEG-speech dataset that is specifically designed for the study of decoding speech from brain signals. The broad applicability of this dataset is crucial for explaining and predicting the neural mechanisms of human language. We not only confirm the quality and completeness of this dataset, but also verify the feasibility of sEEG-based brain-to-speech synthesis. This brain-to-speech synthesis technology provides new research paths at the intersection of neuroscience and artificial intelligence, especially in decoding spoken language, vocabulary categories, frequency bands, and the influence of decoding models.

Although our innovative research and the application of sEEG-speech datasets have demonstrated their obvious advantages, we need to point out some of the negative social impacts they may have. A major problem is that when not all EEG signals can be accurately decoded into understandable speech, this may limit the expression of the patient's true intentions to some extent. Medical staff often need to combine the patient's facial expressions and physiological reactions to more accurately understand their true intentions.

In addition, this technology may have an impact on patients' right to make their own decisions, as they may feel pressured to accept the technology, even though they may have their own concerns. Therefore, we are actively promoting the introduction of more relevant policies to respect and protect patients' right to choose whether to use this technology. We hope that such policies can help ensure the rights and interests of every individual, while providing an important reference for the use of similar technologies in the future.

# 4 Access to Dataset

The CerebroVoice dataset, which is available on Zenodo as a general-purpose open repository, is collected, updated, and maintained by team members from the Big Speech Data Laboratory of The xx. Users can fill out an application form via `https://forms.gle/xkKzYk5KZwZdaSLD9`, upon which the system will immediately and automatically provide a download link for the dataset. The code for dataset creation and experiments can be accessed at `https://github.com/Brain2Speech2/B2S2`.

# 5 Licence

We publish all data under CC-BY-4.0 licence. We include detailed instructions on how to obtain our data and provide preprocessing scripts in our GitHub repository. This dataset is intended for research purposes only and not for clinical usage.

# 6 Implementation Details

## 6.1 Experimental Parameter

In our experiments, to ensure uniformity and fairness across all experimental setups, we applied identical hyperparameter configurations for all comparison tests. Each model was trained over 300 epochs to guarantee convergence in every experiment. Specifically, we set the batch size to 16 and chose an initial learning rate of 0.0625. Utilizing the Adam optimizer with betas parameters of 0.9 and 0.98 allowed us to regulate the exponential moving average of both the gradient and its squared form, aiming to achieve a balance between training stability and speed. Additionally, we implemented a gradient clipping threshold of 1.0 to effectively mitigate the risk of gradient explosion. Additionally, we implemented a warm-up strategy to stabilize the training process.

## 6.2 Evaluation Metrics

PCC (Pearson Correlation Coefficient) is a statistical indicator used to measure the strength and direction of the linear relationship between two variables. PCC is the most commonly used metric in the field of sEEG-based speech decoding[1–4]. The value range of this indicator is between -1 and 1, where:

- If PCC is equal to 1, it means that the two variables are completely positively correlated, that is, when one variable increases, the other variable also increases, and the relationship between the two is linear.

- If PCC is equal to -1, it means that the two variables are completely negatively correlated, that is, when one variable increases, the other variable decreases, which is also a linear relationship.
- If PCC is equal to 0, it means that there is no linear relationship between the two variables.

# 7 Authorstatement

As the authors, we solemnly assure that we accept full responsibility for any possible infringements regarding the data compilation or related proceedings, and commit to promptly taking necessary steps - such as data removal - when dealing with such issues.

# 8 Information Sheet and Consent Form of Participants

In the following sections, we provide a detailed overview of the Consent Agreement and the Experiment Research Information Sheet. Each participant was required to thoroughly review the Experiment Research Information Sheet before consenting to participate. Upon agreeing to the terms outlined, participants signed the Consent Agreement prior to their involvement in the study.

## 9 The Comprehensive Performance Evaluation of VAD

Table 2: Comprehensive Performance Evaluation of VAD for Subject 1

| sEEG feature | Models | Acc | MR | FAR | ER | Prec | Rec | F1 | BA | AUROC |
|---|---|---|---|---|---|---|---|---|---|---|
| HGA | STANet | 0.722 | 0.070 | 0.208 | 0.278 | 0.245 | 0.490 | 0.326 | 0.624 | 0.684 |
| | EEGNet | 0.728 | 0.060 | 0.212 | 0.272 | 0.269 | 0.566 | 0.365 | 0.660 | 0.722 |
| | ECN | 0.764 | 0.035 | 0.200 | 0.236 | 0.338 | 0.743 | 0.465 | 0.755 | 0.834 |
| LFS | STANet | 0.818 | 0.034 | 0.148 | 0.182 | 0.412 | 0.755 | 0.533 | 0.792 | 0.856 |
| | EEGNet | 0.813 | 0.033 | 0.154 | 0.187 | 0.405 | 0.764 | 0.530 | 0.792 | 0.852 |
| | ECN | 0.868 | 0.037 | 0.095 | 0.132 | 0.515 | 0.732 | 0.605 | 0.811 | 0.905 |
| BBS | STANet | 0.801 | 0.049 | 0.150 | 0.199 | 0.371 | 0.644 | 0.471 | 0.735 | 0.806 |
| | EEGNet | 0.813 | 0.028 | 0.159 | 0.187 | 0.409 | 0.797 | 0.540 | 0.807 | 0.867 |
| | ECN | 0.876 | 0.026 | 0.098 | 0.124 | 0.532 | 0.814 | 0.644 | 0.850 | 0.928 |

**Note:** Acc: Accuracy, MR: Miss Rate, FAR: False Alarm Rate, ER: Error Rate, Prec: Precision, Rec: Recall, F1: F1 Score, BA: Balanced Accuracy, AUROC: Area Under the Receiver Operating Characteristic Curve, ECN: EEGChannelNet

Table 3: Comprehensive Performance Evaluation of VAD for Subject 2

| sEEG feature | Models | Acc | MR | FAR | ER | Prec | Rec | F1 | BA | AUROC |
|---|---|---|---|---|---|---|---|---|---|---|
| HGA | STANet | 0.576 | 0.073 | 0.351 | 0.424 | 0.239 | 0.604 | 0.343 | 0.587 | 0.622 |
| | EEGNet | 0.509 | 0.052 | 0.439 | 0.491 | 0.230 | 0.715 | 0.348 | 0.589 | 0.620 |
| | ECN | 0.546 | 0.045 | 0.409 | 0.454 | 0.252 | 0.752 | 0.377 | 0.626 | 0.675 |
| LFS | STANet | 0.584 | 0.044 | 0.371 | 0.416 | 0.272 | 0.757 | 0.400 | 0.651 | 0.699 |
| | EEGNet | 0.595 | 0.043 | 0.362 | 0.405 | 0.278 | 0.763 | 0.408 | 0.660 | 0.712 |
| | ECN | 0.618 | 0.038 | 0.344 | 0.382 | 0.296 | 0.790 | 0.430 | 0.684 | 0.752 |
| BBS | STANet | 0.629 | 0.060 | 0.311 | 0.371 | 0.284 | 0.673 | 0.399 | 0.646 | 0.695 |
| | EEGNet | 0.639 | 0.051 | 0.311 | 0.361 | 0.299 | 0.723 | 0.423 | 0.672 | 0.724 |
| | ECN | 0.666 | 0.031 | 0.303 | 0.334 | 0.334 | 0.831 | 0.476 | 0.730 | 0.803 |

**Note:** Acc: Accuracy, MR: Miss Rate, FAR: False Alarm Rate, ER: Error Rate, Prec: Precision, Rec: Recall, F1: F1 Score, BA: Balanced Accuracy, AUROC: Area Under the Receiver Operating Characteristic Curve, ECN: EEGChannelNet

**Accuracy (Acc):** The proportion of correctly identified instances (both true positives and true negatives) over the total number of instances. It provides an overall measure of the model's performance.

$$\text{Accuracy} = \frac{\text{TP} + \text{TN}}{\text{TP} + \text{TN} + \text{FP} + \text{FN}} \tag{1}$$

**Miss Rate (MR):** The proportion of actual positive instances (events where the subject is speaking) that are incorrectly identified as negative (missed). It is also known as the false negative rate.

$$\text{Miss Rate} = \frac{\text{FN}}{\text{TP} + \text{TN} + \text{FP} + \text{FN}} \tag{2}$$

**False Alarm Rate (FAR):** The proportion of actual negative instances (events where the subject is not speaking) that are incorrectly identified as positive (false alarms). It is also known as the false positive rate.

$$\text{False Alarm Rate} = \frac{\text{FP}}{\text{TP} + \text{TN} + \text{FP} + \text{FN}} \tag{3}$$

**Error Rate (ER):** The proportion of all instances that are incorrectly classified. This includes both false positives and false negatives.

$$\text{Error Rate} = \frac{\text{FP} + \text{FN}}{\text{TP} + \text{TN} + \text{FP} + \text{FN}} \tag{4}$$

**Precision (Prec):** The proportion of predicted positive instances that are correctly identified. It indicates the accuracy of the positive predictions.

$$\text{Precision} = \frac{\text{TP}}{\text{TP} + \text{FP}} \tag{5}$$

**Recall (Rec):** The proportion of actual positive instances that are correctly identified. It is also known as sensitivity or true positive rate.

$$\text{Recall} = \frac{\text{TP}}{\text{TP} + \text{FN}} \tag{6}$$

**F1 Score (F1):** The harmonic mean of precision and recall, providing a single measure that balances both concerns.

$$\text{F1 Score} = 2 \times \frac{\text{Precision} \times \text{Recall}}{\text{Precision} + \text{Recall}} \tag{7}$$

**Balanced Accuracy (BA):** The average of the true positive rate and the true negative rate. It accounts for class imbalance by considering both recall of the positive and negative classes.

$$\text{Balanced Accuracy} = \frac{\text{Recall} + \text{Specificity}}{2} \tag{8}$$

**Area Under the Receiver Operating Characteristic Curve (AUROC):** A measure of the model's ability to discriminate between positive and negative classes. It plots the true positive rate against the false positive rate at various threshold settings.

$$\text{AUROC} = \int_0^1 \text{TPR}(\text{FPR}) \, d(\text{FPR}) \tag{9}$$

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
