# OpenReview forum: "CerebroVoice: A Stereotactic EEG Dataset and Benchmark for Bilingual Brain-to-Speech Synthesis and Activity Detection"
_ICLR.cc/2025/Conference — Submitted to ICLR 2025_

### Official Review · Reviewer_u8GS · 2024-10-27

**Soundness:** 2
**Presentation:** 3
**Contribution:** 3
**Rating:** 6
**Confidence:** 3

**Summary:**

The authors introduce a novel dataset, CerebroVoice (publicly available), for bilingual brain-to-speech synthesis and a neural architecture MoBSE, which utilizes a language-aware prior dynamic organization for efficient handling of language-specific decoding tasks.

**Dataset**: The audio stimulus set contains `50` different stimuli, including 30 Chinese Mandarin words, 10 Chinese Mandarin digits, and 10 English words. For each trial, one randomly selected audio stimulus is played; then, the patient is asked to repeat that word (or digit). The dataset includes `1600` trials (i.e., 29 trials per Chinese Mandarin word, 48 trials per Chinese Mandarin digit, and 24 trials per English word). In each trial, two kinds of brain responses are recorded, including listening and reading. Each trial lasts either `4` or `5` seconds and is paired with the corresponding audio recording.

**Model**: The authors propose MoBSE, which is similar to `model ensemble`. MoBSE uses an additional gating module to support the dynamical fusion of the outputs from different experts.

**Experiment**: Previous methods (e.g., FastSpeech2, EEGNet, STANet, EEGChannelNet) are compared. Besides, the authors conducted different ablation studies regarding sEEG settings (sEEG feature, subject, word categories, etc.).

**In summary, it seems like a dataset paper.**

-----------

**Summary**

I have throughly seen other reviewers' comments, I decide to decrease my score to 6. **This article is overall at a borderline level, and the author may consider collecting more data to further enhance the manuscript.**

 - the limited qualitative assessment, an extremely small number of participants in the dataset (only 3 subjects), compared to NeurIPS 2024 dataset paper Brain Treebank [1].
 - the lack of a detailed analysis based on existing datasets to demonstrate the value added from the new dataset (e.g., detailed distribution analysis and cross-dataset testing), compared to Edward Chang's NBE paper [2].

This work provides a sEEG alternative to ECoG-based bilingual speech dataset [2].

**Reference**:

[1] Wang C, Yaari A U, Singh A K, et al. Brain Treebank: Large-scale intracranial recordings from naturalistic language stimuli[J]. arXiv preprint arXiv:2411.08343, 2024.

[2] Silva A B, Liu J R, Metzger S L, et al. A bilingual speech neuroprosthesis driven by cortical articulatory representations shared between languages[J]. Nature Biomedical Engineering, 2024: 1-15.

**Strengths:**

**Significance**: Open-source sEEG speech datasets are rare. Their publishing of the dataset (Line 035) is good news for the community as it will lower the entry threshold for future research. Additionally, they demonstrate how different sEEG features (e.g., LFS, HGA, BBS) affect the performance of brain-to-speech synthesis and voice activity detection. These results may help future works on speech decoding.

**Clarity**: The text has a good structure and is well-written. The figures also help in understanding the method.

**Weaknesses:**

**Major**
1. Why is common average referencing, instead of laplacian reference, used in BrainBERT[1] (for listening decoding) or bipolar reference used in Du-IN[2] (for speech decoding)? Could you provide brain-to-speech synthesis results based on either laplacian reference or bipolar reference? Although previous studies[3] on speech synthesis use common average referencing + HGA, the speech synthesis task has a trivial solution (the mel-spectrum distribution of human speech is easy to regress). Maybe I’m wrong, but with these additional results, we can gain a deeper understanding of the dataset. Could the authors include the results of brain-to-speech synthesis (i.e., Table 1) baesd on the preprocessed data after either laplacian reference or bipolar reference?

2. How about the results of word classification? CerebroVoice dataset includes at least `24` trials per words, it should be able to evaluate 30-way classification task (i.e., 30 Chinese Mandarin words). Could the authors include results on word-classification tasks (e.g., 30-way on Chinese words, 10-way on Chinese digits, 10-way on English words)?

**Minor**
1. Line 90: Additional publications the authors should be aware:
  - In Du-IN (https://arxiv.org/abs/2405.11459), their preprocessed dataset is open available.

Could the authors summarize these works in Table 1?

2. Line 99: Additional publications the authors should be aware:
  - In Feng et al. (https://www.biorxiv.org/content/10.1101/2023.11.05.562313v3), they also explore speech decoding based on tonal language (i.e., Chinese Mandarin).

Could the authors summarize these works in the Related Works?

**Reference**

[1] Wang C, Subramaniam V, Yaari A U, et al. BrainBERT: Self-supervised representation learning for intracranial recordings[J]. arXiv preprint arXiv:2302.14367, 2023.

[2] Zheng H, Wang H T, Jiang W B, et al. Du-IN: Discrete units-guided mask modeling for decoding speech from Intracranial Neural signals[J]. arXiv preprint arXiv:2405.11459, 2024.

[3] Chen J, Chen X, Wang R, et al. Subject-Agnostic Transformer-Based Neural Speech Decoding from Surface and Depth Electrode Signals[J]. bioRxiv, 2024.

**Questions:**

1. Line 162: What does “a Python-scripted audio playback and sEEG-marking mechanism” mean? At the onset of audio stimuli (not the participant’s audio), the system sends a marker to ths sEEG recordings to identify the onset of audio stimuli.

---

> ### Author Response · Authors · 2024-11-25
> **Response to Reviewer u8GS: Q1 & Q2 & Q3 & Q4**
>
> **Q1: Why is common average referencing, instead of laplacian reference, used in BrainBERT[1] (for listening decoding) or bipolar reference used in Du-IN[2] (for speech decoding)? Could you provide brain-to-speech synthesis results based on either laplacian reference or bipolar reference? Although previous studies[3] on speech synthesis use common average referencing + HGA, the speech synthesis task has a trivial solution (the mel-spectrum distribution of human speech is easy to regress). Maybe I’m wrong, but with these additional results, we can gain a deeper understanding of the dataset. Could the authors include the results of brain-to-speech synthesis (i.e., Table 1) based on the preprocessed data after either laplacian reference or bipolar reference?**
>
> **Response:** Thank you for your insightful suggestions. We initially selected common average referencing due to its common use and effectiveness in preprocessing brain signals. As suggested, we explored the use of bipolar referencing, which emphasizes differences between adjacent electrodes.
>
> The detailed results, summarized in Table 3, indicate that the performance with bipolar referencing closely aligns with that of common average referencing. Specifically, there were no significant changes were observed in the synthesis quality between the two approaches. However, we agree with the reviewer’s perspective that bipolar referencing offers a potentially more localized perspective on neural signals. In light of this, we will update all results obtained using bipolar referencing accordingly.
>
> **Table 3 Speech synthesis performance across different spoken word categories and sEEG features with bipolar referencing for Subject 1 using MoBSE.**
>
> | sEEG Feature | CN   | EN   | DIGIT |
> |--------------|------|------|-------|
> | LFS          | 0.638| 0.531| 0.615 |
> | HGA          | 0.642| 0.513| 0.599 |
> | BBS          | 0.673| 0.537| 0.602 |
>
> **Q2: 1.Line 90: Additional publications the authors should be aware: In Du-IN[https://arxiv.org/abs/2405.11459], their preprocessed dataset is open available. Could the authors summarize these works in Table 1?**
>
> **Response:** We appreciate your valuable input. As suggested, we have summarized the key aspects of the Du-IN work in Table 1. Specifically, the Du-IN work focuses on a classification task using sEEG signals, which differs from with our focus on Brain-to-Speech synthesis. We will further clarify this in the revised manuscript.
>
> **Q3: 2.Line 99: Additional publications the authors should be aware: In Feng et al. (https://www.biorxiv.org/content/10.1101/2023.11.05.562313v3), they also explore speech decoding based on tonal language (i.e., Chinese Mandarin). Could the authors summarize these works in the Related Works?**
>
> **Response:** Thank you for your suggestion. We have properly cited these two works in the Related Works section of the updated manuscript.
>
> **Q4: Line 162: What does “a Python-scripted audio playback and sEEG-marking mechanism” mean? At the onset of audio stimuli (not the participant’s audio), the system sends a marker to the sEEG recordings to identify the onset of audio stimuli.**
>
> **Response:** Thank you for your comment. The reviewer is correct that "a Python-scripted audio playback and sEEG-marking mechanism" refers to a system implemented in Python that plays audio stimuli and simultaneously sends a marker to the sEEG recordings at the onset of the audio. We have revised this sentence to clarify its meaning.
>
> ***"To ensure synchronization between the auditory stimuli and sEEG responses, we employed a Python-scripted tool to play audio stimuli and simultaneously mark the corresponding sEEG responses."***
>
> ***Reference***
>
> [1] Wang C, Subramaniam V, Yaari A U, et al. BrainBERT: Self-supervised representation learning for intracranial recordings[J]. arXiv preprint arXiv:2302.14367, 2023.
>
> [2] Zheng H, Wang H T, Jiang W B, et al. Du-IN: Discrete units-guided mask modeling for decoding speech from Intracranial Neural signals[J]. arXiv preprint arXiv:2405.11459, 2024.
>
> [3] Chen J, Chen X, Wang R, et al. Subject-Agnostic Transformer-Based Neural Speech Decoding from Surface and Depth Electrode Signals[J]. bioRxiv, 2024.

---

> > ### Comment · Reviewer_u8GS · 2024-11-25
> > **Official Comment by Reviewer u8GS**
> >
> > Thank you for the responses. These results help.
> >
> > > However, we agree with the reviewer’s perspective that bipolar referencing offers a potentially more localized perspective on neural signals.
> >
> > The bi-polar reference might perform better for certain classification tasks without continuous labels (e.g., BrainBERT [1], Du-IN [2], Seegnificant [3]), as it can enhance the prominence of feature waves produced by local neuronal groups.
> >
> > When the decoding system is applied to real patients, such as those with ALS, the exact moment the patient begins speaking is often uncertain. In such cases, using a bipolar reference can lead to greater performance improvements.
> >
> > **Summary**:
> >
> > Thanks for your contribution to the open-source sEEG dataset. This innovative sEEG experimental paradigm holds significant research value. **I have raised my score to 8, conditional on all ablation results (in Q1) being added to the paper.** Good luck :)
> >
> > **Reference**:
> >
> > [1] Wang C, Subramaniam V, Yaari A U, et al. BrainBERT: Self-supervised representation learning for intracranial recordings[J]. arXiv preprint arXiv:2302.14367, 2023.
> >
> > [2] Zheng H, Wang H T, Jiang W B, et al. Du-IN: Discrete units-guided mask modeling for decoding speech from Intracranial Neural signals[J]. arXiv preprint arXiv:2405.11459, 2024.
> >
> > [3] Mentzelopoulos G, Chatzipantazis E, Ramayya A G, et al. Neural decoding from stereotactic EEG: accounting for electrode variability across subjects[C]//The Thirty-eighth Annual Conference on Neural Information Processing Systems.

---

> > > ### Author Response · Authors · 2024-11-25
> > > **Thank You, Reviewer u8GS, for the Revised Rating and Feedback**
> > >
> > > **We would like to express our gratitude for your reassessment of our work and for raising your rating to 8！ We deeply appreciate your careful consideration of our paper and the valuable feedback you have shared during the review process.**

---

### Official Review · Reviewer_M7cJ · 2024-10-29

**Soundness:** 3
**Presentation:** 4
**Contribution:** 2
**Rating:** 5
**Confidence:** 4

**Summary:**

The paper introduces CerebroVoice, a new dataset for bilingual brain-to-speech synthesis and Voice Activity Detection (VAD) using stereotactic EEG (sEEG). It includes recordings from two bilingual participants who read Chinese Mandarin words, English words, and Chinese Mandarin digits. The authors developed a novel method called Mixture of Bilingual Synergy Experts (MoBSE) that uses a language-aware dynamic organization of low-rank expert weights and tested it against the FastSpeech2 baseline, setting a new benchmark for their dataset. They found that MoBSE performs better than FastSpeech2 in producing speech from neural recordings. Additionally, they reproduced three existing VAD methods and established benchmarks for VAD using CerebroVoice. The dataset is publicly available on Zenodo, and the preprocessing code can be found on GitHub.

**Strengths:**

-The authors introduce CerebroVoice, a publicly accessible sEEG dataset tailored for neural to speech synthesis and Voice Activity Detection (VAD). This is particularly significant given the scarcity of publicly available sEEG datasets and benchmarks, providing a valuable resource for researchers to compare and validate their methods, fostering progress in brain-computer interface applications.

-By incorporating bilingual data, specifically focusing on a tonal language, Chinese Mandarin, the dataset opens new avenues for research, addressing the complexities associated with tonal languages in brain-to-speech synthesis.

-The methodology for data acquisition is thoroughly and clearly explained, ensuring transparency.

-The authors introduce a Mixture of Experts (MoE)-based framework for neural-to-speech synthesis, which improves bilingual decoding by dynamically organizing language-specific experts. This novel approach outperforms the FastSpeech2 baseline, demonstrating its effectiveness.

-The authors address important ethical concerns related to patient privacy and the sensitive nature of invasive neural recordings, demonstrating a strong commitment to ethical research practices.

**Weaknesses:**

-The CerebroVoice dataset is limited by its small size, featuring only two participants and a repetitive, narrow vocabulary, which restricts its generalizability and raises concerns about potential overfitting. Its focus on simple speech synthesis tasks diminishes its flexibility for broader neuroscience research areas such as brain decoding and semantic reconstruction. Additionally, the task design lacks originality, as many similar speech synthesis/reconstruction objectives have been addressed in previous studies [1, 2, 3], Most existing invasive datasets can be requested from the authors while non-invasive ones are generally publicly available, reducing the novelty of CerebroVoice’s contribution to the field. To enhance its impact, the authors could consider expanding the dataset with more participants and a more diverse vocabulary and/or task in future work.

-The GitHub repository lacks implementations of the proposed models, hindering reproducibility and preventing other researchers from building upon the work. It would be beneficial for the authors to include model implementations, training scripts, and detailed documentation in their GitHub repository.

 -It is unclear how FastSpeech2 was adapted to produce audio from sEEG signals. The paper does not provide a detailed explanation of the training procedures, architectural changes, or loss functions used in adapting this text-to-speech model for brain-to-speech synthesis. Providing specific details about these adaptations would make the methodology more understandable and reproducible.

-The architecture of the experts within the MoBSE framework is not clearly explained, leaving gaps in understanding how the model functions. It does not specify how many experts were used in the MoBSE framework and lacks ablation studies to justify this choice, hindering the evaluation of the model's components.

-The evaluation primarily uses Pearson Correlation Coefficient (PCC). Including additional metrics like ESTOI (Extended Short-Time Objective Intelligibility) would provide a more comprehensive assessment of speech synthesis quality. This is a very common metric in speech synthesis/reconstruction tasks.

[1] M. Angrick, M. Ottenhoff, S. Goulis, A. J. Colon, L. Wagner, D. J. Krusienski, P. L. Kubben,
T. Schultz, and C. Herff, “Speech synthesis from stereotactic EEG using an electrode shaft dependent multi-input convolutional neural network approach,” in 2021 43rd Annual International Conference of the IEEE Engineering in Medicine & Biology Society (EMBC)

[2] Verwoert, Maxime, et al. "Dataset of speech production in intracranial electroencephalography." Scientific data 9.1 (2022): 434.

[3] Akbari, Hassan, et al. "Towards reconstructing intelligible speech from the human auditory cortex." Scientific reports 9.1 (2019): 874.

**Questions:**

Did you perform any statistical significance testing to confirm that the improvements of MoBSE over FastSpeech2 are meaningful?

Is there a reason why raw sEEG data is not provided alongside the processed data, allowing researchers to perform custom preprocessing and explore different frequency bands?

How and why is positional encoding used in the MoBSE framework? Can you provide more insight into its implementation?

Are there any samples of the reconstructed speech available for qualitative assessment?

How is VAD accuracy measured exactly? I'm trying to figure out if you chose a window of silence vs speech? how long was the window?

Have you considered combining electrode data from both subjects to create a "super subject" to enhance coverage?

Thanks,

**Details Of Ethics Concerns:**

While the authors mention addressing ethical concerns related to patient privacy, I still believe it should be reviewed by the ethics committee, as it is very sensitive to make invasive human neural data publicly available. There is insufficient discussion on data storage security measures, access controls, and compliance with data protection regulations such as GDPR.

---

> ### Author Response · Authors · 2024-11-25
> **Response to Reviewer M7cJ: Q1 & Q2**
>
> **Q1: Most existing invasive datasets can be requested from the authors while non-invasive ones are generally publicly available, reducing the novelty of CerebroVoice’s contribution to the field. To enhance its impact, the authors could consider expanding the dataset with more participants and a more diverse vocabulary and/or task in future work.**
>
> **Response:** Thank you for your constructive feedback. We appreciate your thoughtful observations, as the challenges you raised are indeed significant in this field.
>
> As mentioned in the paper, we compared our dataset with previous studies [2], **highlighting that existing datasets are often limited in size and typically focus on a single language.** Additionally, some datasets, **such as [1], are proprietary and not easily accessible to the public, which further constrain research opportunities.**
>
> We agree that brain decoding and semantic reconstruction are intriguing tasks, and high-quality speech synthesis is a key prerequisite. Indeed, speech synthesis and reconstruction remain active areas of research [1,2,3], especially in the context of speech BCIs [4]. We believe that our dataset will continue to be a valuable resource in advancing these areas.
>
> **It is important to note that our dataset is continuously expanding. We are currently collecting data from a new subject [https://zenodo.org/records/14179222]** and are committed to broadening its scope for future research.
>
> **Additionally, our data can be downloaded directly without any request, simply by agreeing to a usage agreement, which is intended to prevent non-academic use.**
>
> We believe that the unique contributions of our dataset, particularly its ongoing expansion and its focus on integrating deep brain signals for speech synthesis, will continue to enhance its value to the field.
>
> ***References***
>
> [1] M. Angrick, M. Ottenhoff, S. Goulis, A. J. Colon, L. Wagner, D. J. Krusienski, P. L. Kubben, T. Schultz, and C. Herff, “Speech synthesis from stereotactic EEG using an electrode shaft dependent multi-input convolutional neural network approach,” in 2021 43rd Annual International Conference of the IEEE Engineering in Medicine & Biology Society (EMBC) .
>
> [2] Verwoert, Maxime, et al. "Dataset of speech production in intracranial electroencephalography." Scientific data 9.1 (2022): 434.
>
> [3] Akbari, Hassan, et al. "Towards reconstructing intelligible speech from the human auditory cortex." Scientific reports 9.1 (2019): 874.
>
> [4] Silva, A. B., Littlejohn, K. T., Liu, J. R., Moses, D. A., & Chang, E. F. (2024). The speech neuroprosthesis. Nature Reviews Neuroscience, 1-20.
>
> **Q2: It would be beneficial for the authors to include model implementations, training scripts, and detailed documentation in their GitHub repository.**
>
> **Response:** Thank you for your valuable feedback. As suggested, we have included the implementations of the models and checkpoints, along with data preprocessing scripts, evaluation metric scripts, and a README file to explain how these components work together. Additionally, we have ensured that the CerebroVoice dataset is easily accessible to the research community.
>
> The dataset can be accesed through the following links:
>
> **For the newly added participant data: https://zenodo.org/records/14179222.**
>
> **For the sEEG data of all subjects: https://zenodo.org/records/13332808.**
>
> We appreciate your suggestion and the opportunity to enhance our repository.

---

> > ### Author Response · Authors · 2024-11-25
> > **Response to Reviewer M7cJ: Q3 & Q4**
> >
> > **Q3: It is unclear how FastSpeech2 was adapted to produce audio from sEEG signals.**
> >
> > **Response:** Thank you for your insightful comment. We appreciate the opportunity to present the adaptation of FastSpeech2 for brain-to-speech synthesis. Below, we provide a detailed explanation of the adaptations, training procedures, architectural modifications, and loss functions employed in our study.
> >
> > Architecture modifications: In the original FastSpeech2 [1], text embeddings are used as input to the encoder. For our sEEG-based speech synthesis task, we replaced these text embeddings with embeddings derived from sEEG signals. Specifically, we transformed 1.5-second sEEG signals into a 2D data format with dimensions (75, 228), where 75 represents the time dimension and 228 represents the channel dimension. This transformation is analogous to the mel-spectrogram features, which have dimensions of (75, 80) in which 80 is the dimension of features and each frame lasts 0.02-second, ensuring alignment with the temporal structure of the speech.  Furthermore,  we introduced the Mixture of Bilingual Synergy Experts (MoBSE) component within the feedforward network (FFN) to effectively handle bilingual decoding tasks, as detailed in Section 5.2.
> >
> > **Training Procedures:** We adapted the training methodology to suit our task requirements. The Adam optimizer was employed with hyperparameters β₁ = 0.9 and β₂ = 0.98. The model was trained with a batch size of 16 and a learning rate of 0.001.
> >
> > **Loss Functions:** The L1 loss was used to measure the difference between the predicted and ground-truth mel-spectrograms.
> >
> > We will revise the manuscript to include these details. Thank you again for your valuable feedback.
> >
> > ***References***
> >
> > [1]Ren Y, Hu C, Tan X, et al. Fastspeech 2: Fast and high-quality end-to-end text to speech[J]. arXiv preprint arXiv:2006.04558, 2020.
> >
> > **Q4: It does not specify how many experts were used in the MoBSE framework and lacks ablation studies to justify this choice, hindering the evaluation of the model's components.**
> >
> > **Response:** Thank you for your feedback.  We chose to use 8 experts in the MoBSE framework based on results from ablation studies, which tested configurations with 4, 6, 8, 10, and 12 experts. Taking BBS features as an example, the average Pearson correlation coefficients (averaged across Chinese, English, and digits) for Subject 1 were 0.589, 0.598, 0.612, 0.607, and 0.602 for configurations with 4, 6, 8, 10, and 12 experts, respectively. For Subject 2, the corresponding values were 0.428, 0.437, 0.453, 0.451, and 0.448. Based on these results, the configuration with 8 experts achieved the best overall performance,striking a balance between effective language-specific decoding and minimizing redundancy or overfitting. We will include the detailed ablation study results in the revised manuscript. Thank you again for your feedback.

---

> > > ### Author Response · Authors · 2024-11-25
> > > **Response to Reviewer M7cJ: Q5**
> > >
> > > **Q5: Incorporating a broader range of qualitative and quantitative evaluation metrics, both subjective and objective, will enhance the comprehensive assessment of the speech synthesis quality.**
> > >
> > > **Response:** Thank you for your valuable feedback. We appreciate the suggestion to incorporate additional evaluation metrics to provide a more comprehensive assessment of speech synthesis quality. We have included the Short-Time Objective Intelligibility (STOI) metric in our evaluation. Furthermore, we expanded our evaluation to include both subjective and objective metrics, including the Mean Opinion Score (MOS), Mel Cepstral Distortion (MCD), Root Mean Squared Error (RMSE), as well as NISQA.
> > >
> > > ***No.1*** Subjective Evaluation: We conducted a Mean Opinion Score (MOS) test to assess the quality of the reconstructed speech. Several samples were randomly selected from our dataset, as well as from the NMI-24 [2] and SD-22 [3] demos, and assessed by 19 raters on a scale from 1 to 10, in increments of 1 point. The test focused on two aspects:
> > >
> > > **●Naturalness: How natural and lifelike does the speech sound?**
> > >
> > > **●Intelligibility: How clearly can you understand the spoken content?**
> > >
> > > Our speech demo achieved an average score of 7.36 for Naturalness and 8.15 for Intelligibility. In comparison, NMI-24's average scores were 5.05 for both Naturalness and Intelligibility, while SD-22 scored 1.15 for Naturalness and 2.10 for Intelligibility.
> > >
> > > The survey used to obtain the MOS scores is available at the following link: https://wj.qq.com/s2/16471941/ac0a/.
> > >
> > > **Objective Evaluation:**
> > >
> > > ***No.2*** **Mel Cepstral Distortion (MCD):** Our model achieved an MCD of 4.143 dB on the CerebroVoice test set, samller than the result of 5.64 dB from BrainTalker [4]. Lower MCD values indicate better performance, suggesting that our model obtains more accurate spectral representations.
> > >
> > > ***No.3*** **Root Mean Squared Error (RMSE):** The RMSE for our model was 0.501 on the CerebroVoice test set, outperforming BrainTalker's best result of 1.28 [4]. Lower RMSE values indicate better performance, further confirming the enhanced quality of our synthesized speech.
> > >
> > > ***No.4*** **Short-Time Objective Intelligibility (STOI):** The average STOI score for our model was 0.2852 on the CerebroVoice test set. As pointed out in your feedback, the STOI metric is crucial for assessing intelligibility in synthesized speech, validating the effectiveness of our approach.
> > >
> > > ***No.5*** Additionally, **we incorporated the commonly used no-reference speech quality assessment metric, NISQA (cited 229 times) [1]**, which **evaluates the quality of generated human speech without requiring a groundtruth reference**. We applied NISQA to our decoded speech samples and also evaluated publicly available speech samples from **Nature Machine Intelligence** (NMI-24, 2024) [2] and **Scientific Data** (SD-22, 2022) [3], with higher scores reflecting better quality. Our decoded speech achieved a score of 3.2751, outperforming the results from [2] and [3], which scored 2.2828 and 1.8911, respectively.
> > >
> > > These results validate the effectiveness of our approach, providing a comprehensive comparison with existing methods and datasets. We will inclued these  evaluation metrics in the revised version. We greatly appreciate your insightful suggestions, which have significantly enhanced the quality  of our study.
> > >
> > > ***References***
> > >
> > > [1]Mittag G, Naderi B, Chehadi A, et al. NISQA: A deep CNN-self-attention model for multidimensional speech quality prediction with crowdsourced datasets[J]. arXiv preprint arXiv:2104.09494, 2021.
> > >
> > > [2] Chen X, Wang R, Khalilian-Gourtani A, et al. A neural speech decoding framework leveraging deep learning and speech synthesis[J]. Nature Machine Intelligence, 2024: 1-14.
> > >
> > > [3] Verwoert M, Ottenhoff M C, Goulis S, et al. Dataset of speech production in intracranial electroencephalography[J]. Scientific Data, 2022, 9(1): 434.
> > >
> > > [4] Kim, Miseul, Zhenyu Piao, Jihyun Lee, and Hong-Goo Kang. "BrainTalker: Low-Resource Brain-to-Speech Synthesis with Transfer Learning using Wav2Vec 2.0." In 2023 IEEE EMBS International Conference on Biomedical and Health Informatics (BHI), pp. 1-5. IEEE, 2023.

---

> > > > ### Author Response · Authors · 2024-11-25
> > > > **Response to Reviewer M7cJ: Q6 & Q7 & Q8 & Q9 & Q10 & Q11**
> > > >
> > > > **Q6: Did you perform any statistical significance testing to confirm that the improvements of MoBSE over FastSpeech2 are meaningful?**
> > > >
> > > > **Response:** Thank you for pointing it out. We have performed a paired t-test to confirm the improvements of MoBSE over FastSpeech2. The results of this statistical significance testing have been added in the updated manuscript.
> > > >
> > > > **Q7: Is there a reason why raw sEEG data is not provided alongside the processed data, allowing researchers to perform custom preprocessing and explore different frequency bands?**
> > > >
> > > > **Response:** Thank you for your question. In this version, we provided only the preprocessed sEEG data for clarity and ease of use. However, we agree with the reviewer on the value of raw data for custom preprocessing and exploration of various frequency bands. We will make the raw data available after acceptance of the manuscript.
> > > >
> > > > **Q8: How and why is positional encoding used in the MoBSE framework? Can you provide more insight into its implementation?**
> > > >
> > > > **Response:** Thank you for your question. In the MoBSE framework, we use absolute positional encoding with learnable parameters to capture both temporal and spatial information from sEEG signals. Specifically, we initialize learnable embedding matrices for the temporal dimension (T=75) and the channel dimension (C=228), where each matrix has dimension according to the  respective size and embedding dimension (d). During preprocessing, these positional embeddings are added to the sEEG feature embeddings, enriching them with absolute temporal and spatial context. These embeddings are trainable and are updated during training, enabling the model to optimize them based on the specific task.
> > > >
> > > > Temporal and channel information from sEEG signals are crucial for accurate speech decoding, as they reflect both dynamic changes over time and the contributions of different brain regions. We use absolute positional encoding to provide structural information about the temporal and spatial arrangement of the sEEG signals. Unlike fixed sinusoidal encodings, learnable positional embeddings can  dynamically adapt during training to better represent the unique characteristics of the sEEG data. This approach ensures that the model can effectively capture the order of temporal slices and the relationships between sEEG channels.
> > > >
> > > > **Q9: Are there any samples of the reconstructed speech available for qualitative assessment?**
> > > >
> > > > **Response:** Thank you for your question. We have provided samples of the reconstructed speech for qualitative assessment, which can be accessed via the Zenodo link [https://zenodo.org/records/13332808].
> > > >
> > > > Additionally, for your convenience, the samples are also uploaded in the demo section on our Google Drive [https://drive.google.com/file/d/1QVhNrN-2kOb-paVbkK97wZGHLWlToj4Z/view?usp=sharing].
> > > >
> > > > **Q10: How is VAD accuracy measured exactly? I'm trying to figure out if you chose a window of silence vs speech? how long was the window?**
> > > >
> > > > **Response:** Thank you for your question. You are correct that VAD is used to distinguish between speech and silence within a window. In our study, the window length was set to 0.064 seconds. We will make this clear in the revised version.
> > > >
> > > > **Q11: Have you considered combining electrode data from both subjects to create a "super subject" to enhance coverage?**
> > > >
> > > > **Response:** Thank you for constructive comment. As suggested, we combined the electrode data from both subjects to form a super subject to enhance the brain coverage. Specifically, we used the speech listened to by both subjects as the ground truth for generating the corresponding mel-spectrograms, because using the speech from any single subject would be unfair.
> > > >
> > > > The detailed results are presented in the following Table:
> > > >
> > > > **Table 2 Speech synthesis performance across different spoken word categories and sEEG features with bipolar referencing for Super Subject using MoBSE**
> > > >
> > > > | sEEG Feature | CN   | EN   | DIGIT | Avg  |
> > > > |--------------|------|------|-------|------|
> > > > | LFS          | 0.611| 0.570| 0.580 | 0.587|
> > > > | HGA          | 0.567| 0.544| 0.501 | 0.537|
> > > > | BBS          | 0.626| 0.606| 0.595 | 0.609|
> > > >
> > > >
> > > > We appreciate your suggestion to integrate data across subjects, as it will offer valuable insights and contribute to new perspectives. We will incorporate these results in the updated version.

---

> > > > > ### Author Response · Authors · 2024-11-27
> > > > > **Request for Reevaluation in Light of Our Responses**
> > > > >
> > > > > Dear Reviewer M7cJ,
> > > > >
> > > > > We have compiled all your comments into Q1-Q11 and have provided detailed responses, along with the necessary experiments. **At your convenience, we kindly ask if you could review our responses and reconsider the evaluation of our work.**
> > > > >
> > > > > **We greatly appreciate the time and effort you have dedicated to this process.** If you have any further questions or require additional discussion, we would be more than happy to engage with you further.
> > > > >
> > > > > Thank you for your continued support and guidance.
> > > > >
> > > > > Best regards,
> > > > >
> > > > > Authors

---

> ### Author Response · Authors · 2024-11-25
> **Kindly Request for Re-rating Based on Our Responses**
>
> Dear Reviewer M7cJ,
>
> We sincerely appreciate your review of our manuscript and the valuable feedback you provided. **We highly value your input, which has played a crucial role in refining our paper.** We believe our detailed responses and additional experiments have addressed your concerns.
>
> **If our responses meet your expectations, we kindly request that you consider updating your rating to reflect these improvements.** If there are any issues that remain unresolved, please feel free to contact us at your convenience. Thank you again for your constructive feedback and support.
>
> Best regards,
>
> Authors

---

> ### Author Response · Authors · 2024-11-27
> **Follow-Up: Request for Your Latest Evaluation**
>
> **Dear Reviewer M7cJ,**
>
> We would like to express our sincere gratitude for your thoughtful suggestions and constructive feedback, which have been invaluable in improving our manuscript. **We have carefully considered and addressed all your comments, summarizing them as Q1-Q11, and have provided detailed responses along with additional necessary experiments.**
>
> We kindly ask if you could take some time to review our replies. If you have any further questions or require additional clarifications, please feel free to reach out to us at any time. **Meanwhile, we are eager to engage in further discussions and look forward to receiving your updated evaluation.**
>
> **Thank you once again for your valuable contributions to our work.**
>
> Warm regards,
>
> Authors

---

> ### Author Response · Authors · 2024-11-30
> **Detailed Responses and Revised Manuscript for Your Review**
>
> ***Dear Reviewer M7cJ,***
>
> ***We have carefully addressed your suggestions and organized them into 11 questions, providing detailed responses and supplementing them with necessary experimental content.***
>
> ***
>
> ***
>
> ***For Q1-Q4, Q6-Q8, and Q10-Q11:***
>
> ***We have provided comprehensive responses in the "Messages" section, and these improvements have been incorporated into our revised manuscript. You can view the updated PDF in the OpenReview system.***
>
> ***
>
> ***
>
> ***Regarding Q5: Expanding the Evaluation Metrics(also updated in the revised PDF):***
>
> **Based on your suggestions and those of Reviewer yXuZ**, we have conducted extensive experiments and added further explanations. These updates are reflected in the revised PDF, as detailed below.
>
> In line with your suggestion, we conducted a more extensive baseline comparison and adopted more comprehensive metrics for evaluation.
>
> ***
>
> The baselines include ***Shaft CNN***, ***Hybrid CNN-LSTM***, ***Dynamic GCN-LSTM***, ***FastSpeech 2[4]***, as well as ***BrainTalker[1]***, ***NeuroTalk[2]***, and ***ECoG Decoder[3]***, which you indicated for reference. The evaluation metrics include ***PCC***, ***MCD***, ***RMSE***, and ***STOI***, along with ***MOS*** and ***NISQA***[5]. **You can view the updated version in the PDF located at the top right in OpenReview.**
>
> ***
>
> ***For PCC, MCD, RMSE, and STOI, which compare various state-of-the-art methods on our proposed CerebroVoice dataset, the results are as follows:***
>
> | **Subjects** | **Model**          | **PCC** (↑) | **STOI** (↑) | **MCD** (↓) | **RMSE** (↓) |
> |--------------|--------------------|-------------|--------------|-------------|--------------|
> | Subject 1    | ***BrainTalker[1]***        | 0.584       | 0.193        | 4.282       | 0.523        |
> |              | ***MoBSE (Ours)***        | **0.604**   | **0.285**    | **4.143**   | **0.501**    |
> |              | ***Shaft CNN***           | 0.583       | 0.195        | 4.358       | 0.548        |
> |              | ***Hybrid CNN-LSTM***     | 0.564       | 0.170        | 4.448       | 0.562        |
> |              | ***Dynamic GCN-LSTM***    | 0.551       | 0.153        | 4.556       | 0.583        |
> |              | ***FastSpeech 2[4]***        | 0.578       | 0.182        | 4.206       | 0.518        |
> |              | ***ECoG Decoder[3]***        | 0.569       | 0.176        | 4.406       | 0.530        |
> |              | ***NeuroTalk[2]***           | 0.590       | 0.196        | 4.198       | 0.509        |
> | Subject 2    | ***BrainTalker[1]***    | 0.434       | 0.142        | 5.958       | 0.635        |
> |              | ***MoBSE (Ours)***        | **0.452**   | **0.184**    | **5.652**   | **0.622**    |
> |              | ***Shaft CNN***           | 0.432       | 0.153        | 5.986       | 0.644        |
> |              | ***Hybrid CNN-LSTM***     | 0.424       | 0.126        | 6.124       | 0.656        |
> |              | ***Dynamic GCN-LSTM***    | 0.408       | 0.122        | 6.334       | 0.660        |
> |              | ***FastSpeech 2[4]***     | 0.438       | 0.152        | 5.906       | 0.641        |
> |              | ***ECoG Decoder[3]***     | 0.429       | 0.148        | 5.980       | 0.656        |
> |              | ***NeuroTalk[2]***        | 0.442       | 0.162        | 5.707       | 0.637        |
>
> For Subject 1, MoBSE outperformed other models with the highest PCC of 0.604 and STOI of 0.285, indicating improved correlation and intelligibility of the reconstructed speech. Additionally, MoBSE achieved the lowest MCD of 4.143 and RMSE of 0.501, demonstrating superior accuracy and reduced distortion in speech reconstruction. Similarly, for Subject 2, MoBSE maintained its leading performance with a PCC of 0.452 and a STOI of 0.184, along with the lowest MCD of 5.652 and RMSE of 0.622. ***These results consistently show that MoBSE provides a significant improvement in speech quality and intelligibility compared to other methods.***
>
> The results for NeuroTalk[2] and ECoG Decoder[3] are already presented in the following messages. Additionally, the results will be updated in the main text of the paper under "***Table 3: Comparison of MoBSE with other state-of-the-art methods across different subjects***" once the paper is accepted.
>
> ***
>
> ***

---

> ### Author Response · Authors · 2024-11-30
> **Detailed Responses and Revised Manuscript for Your Review**
>
> ***
>
> ***
>
> ***Regarding Q9: Samples for Qualitative Assessment(also updated in the revised PDF):***
>
> We have included samples of the reconstructed speech for qualitative assessment. ***These can be accessed via the Google Drive link [https://drive.google.com/file/d/1QVhNrN-2kOb-paVbkK97wZGHLWlToj4Z/view?usp=sharing]. Additionally, we have conducted thorough experiments related to Q9, which are also included in the revised PDF.***
>
> ***
>
> ***To more fairly emphasize the high quality of our data, we performed a comprehensive comparison of the speech generated by our CerebroVoice system against the outputs from existing research[6-7].***
>
>
> ***
>
> | **Metric**                | **CerebroVoice** | **NMI-24[6]** | **SD-22[7]** |
> |---------------------------|------------------|------------|-----------|
> | **Mean Opinion Score (MOS) (1-5 scale)** | 4.33             | 2.93       | 1.27      |
> | **NISQA Score**           | 3.2751           | 2.2828     | 1.8911    |
>
>
> In a subjective Mean Opinion Score (MOS) test, using a 1-5 scale, 15 raters evaluated the speech samples based on a combination of naturalness and intelligibility. ***CerebroVoice achieved an average score of 4.33, demonstrating superior performance compared to NMI-24[6], which scored 2.93, and SD-22[7], which scored 1.27.*** These results indicate that CerebroVoice generates speech perceived as both more natural and intelligible.
>
> For the objective evaluation, we utilized the NISQA[5] metric, a no-reference speech quality assessment tool. ***CerebroVoice obtained a score of 3.2751, while NMI-24 and SD-22 scored 2.2828 and 1.8911***, respectively. **The alignment between subjective and objective evaluations highlights the superior quality of speech produced by CerebroVoice compared to existing research. This analysis underscores the advancements in speech quality achieved by our system.**
>
> ***
>
> ***We have fully incorporated your valuable suggestions by expanding our experiments, and you can view our revised content in the updated PDF. We sincerely request that you reconsider your evaluation of this work in light of our efforts.***
>
> ***
>
> ***References:***
>
> **[1]** Kim, Miseul, Zhenyu Piao, Jihyun Lee, and Hong-Goo Kang. "BrainTalker: Low-Resource Brain-to-Speech Synthesis with Transfer Learning using Wav2Vec 2.0." In 2023 IEEE EMBS International Conference on Biomedical and Health Informatics (BHI), pp. 1-5. IEEE, 2023.
>
> **[2]** Lee, Young-Eun, Seo-Hyun Lee, Sang-Ho Kim, and Seong-Whan Lee. "Towards voice reconstruction from EEG during imagined speech." In Proceedings of the AAAI Conference on Artificial Intelligence, vol. 37, no. 5, pp. 6030-6038. 2023.
>
> **[3]** Metzger, Sean L., Kaylo T. Littlejohn, Alexander B. Silva, David A. Moses, Margaret P. Seaton, Ran Wang, Maximilian E. Dougherty et al. "A high-performance neuroprosthesis for speech decoding and avatar control." Nature 620, no. 7976 (2023): 1037-1046.
>
> **[4]** Ren Y, Hu C, Tan X, et al. Fastspeech 2: Fast and high-quality end-to-end text to speech[J]. arXiv preprint arXiv:2006.04558, 2020.
>
> **[5]** Mittag G, Naderi B, Chehadi A, et al. NISQA: A deep CNN-self-attention model for multidimensional speech quality prediction with crowdsourced datasets[J]. arXiv preprint arXiv:2104.09494, 2021.
>
> **[6]** Chen X, Wang R, Khalilian-Gourtani A, et al. A neural speech decoding framework leveraging deep learning and speech synthesis[J]. Nature Machine Intelligence, 2024: 1-14.
>
> **[7]** Verwoert M, Ottenhoff M C, Goulis S, et al. Dataset of speech production in intracranial electroencephalography[J]. Scientific Data, 2022, 9(1): 434.
>
> ***
>
> ***

---

> ### Author Response · Authors · 2024-11-30
> **Invitation to Review Revised Manuscript and Feedback Responses**
>
> ***Dear Reviewer M7cJ,***
>
> ***
>
> ***We appreciate your valuable feedback and have carefully addressed your suggestions.***
>
> ***We sincerely invite you to review our detailed responses in the "Messages" section and the updates in the revised PDF.***
>
> ***We sincerely hope you will consider re-rating our work based on our efforts.***
>
> ***
>
> ***Best regards,***
>
> ***The Authors***

---

> ### Author Response · Authors · 2024-12-01
> **Request for Timely Review of Revised Manuscript Based on Reviewer M7cJ's Feedback.**
>
> ***Request for Timely Review of Revised Manuscript Based on Reviewer M7cJ's Feedback.***
>
> ***
>
> Dear Reviewer **M7cJ**,
>
> We have thoroughly revised and improved our work based on your feedback. **Considering the upcoming deadline, we kindly ask you to review our revised manuscript at your earliest convenience.**
>
> Best,
>
> Authors

---

### Official Review · Reviewer_asg4 · 2024-10-31

**Soundness:** 3
**Presentation:** 3
**Contribution:** 2
**Rating:** 5
**Confidence:** 4

**Summary:**

The paper presents a data set consisting of pairs of stereotactic EEG and speech signals recorded simultaneously and a set of experiments in the context of brain-to-speech synthesis aiming to provide a benchmark for further research in this area. The dataset comprises sEEG and speech signals from two participants, and the protocol included the repetition of auditory stimuli in two languages. The paper also analyses the voice activity detection problem from the sEEG signals. The paper uses a similar architecture to that of the FastSpeech2 TTS model but substitutes phoneme embeddings with a sEEG embedding layer and proposes an alternative way to codify the language information into the network through a MLP layer that weights the feature representation of the network depending on a one-hot-encoding vector that indicates one of two possible languages in the dataset.

**Strengths:**

The paper addresses a relevant topic and introduces a new open dataset that can help advance a field far from being consolidated and where the data is highly costly and complex to acquire.  It also provides relevant measures that can help objectively evaluate the improvement of further approaches in this field.

**Weaknesses:**

The general novelty of the work is limited. The introduced dataset is valuable and constitutes a significant contribution to the academic community because of its complexity, but with such a limited number of participants in the study, it is hard to consider this work a valid benchmark for the task.  Moreover, the proposed mixture of bilingual synergy experts component is not presented clearly, and the whole pipeline is not well presented.

I acknowledge the authors for addressing most of my questions. Still, the paper's main drawback is that the small number of samples is insufficient to support the authors' claims to consider the proposed dataset as a benchmark. Moreover, several results are inconclusive because they come from two different models (one per subject/electrode position), which makes the manuscript's contribution unclear. Therefore, I agree that, in its current state, the global score of the paper is below the acceptance threshold.

**Questions:**

- Why do the authors argue that other datasets can not be used for VAD if the labels for that task are obtained automatically?
- The authors assert that the audio quality was assessed and the recordings edited accordingly during the data curation process. Was this task performed subjectively? Who was in charge of this task?
- Specifications of audio recording equipment were not included, which is relevant to analysis results and prevent biases in case future data fusion tests can be performed.
- The authors presented independent results per subject. Were these results obtained using a single model trained with data from the two subjects, or were also two models trained (one per patient)?
- Results regarding LFS, HGA, and BBS signals are confusing. There is no apparent coherence regarding frequency bands or between subjects' behavior. Why do the authors consider that these experiments provide a benchmark in this field, considering the scarcity of subjects, which limits the power of any analysis?
- The organization of the paper could be improved. The meaning of LFS, HGA, and BBS features and the relevance of their evaluation should be presented in section 5.

---

> ### Author Response · Authors · 2024-11-24
> **Response to Reviewer asg4: Q1 & Q2 & Q3 & Q4 & Q5**
>
> **Q1: Clarification regarding the scale of the CerebroVoice dataset**
>
> **Response:** Thank you for your valuable feedback. We appreciate your recognition of the complexity of our  dataset and its contribution to the academic community.
>
> As pointed by the reviewer, there are few publicly available datasets in this field, and those that do exist are often limited in size. This is indeed a challenge we are actively working to address. We would like to emphasize that the dataset presented in our work is unique due to its bilingual nature, combining both tonal language (Chinese) and non-tonal language (English), which provides unique value to this research area.
>
> In this paper, we report the debut of a significant data collection effort and its formulation in a systematic manner. As we continue to add more participants and data, this dataset will further strengthen its role as a benchmark for shared tasks. The updates can be accessed at https://zenodo.org/records/14179222.
>
> We will revise the manuscript to better highlight these points, as well as our ongoing efforts to expand the dataset, further strengthening its potential impact on the field.
>
> **Q2: A more detailed explanation of the proposed Mixture of Bilingual Synergy Experts(MoBSE) component and the overall pipeline**
>
> **Response:** Thank you for pointing it out. We introduce MoBSE as a language-aware dynamic organization of low-rank expert weights within the feedforward network (FFN) of the Transformer architecture. As outlined in Equations (1)-(3): Equation (1) illustrates how input features and task labels (e.g., Mandarin or English) are fused into a unified representation. Equation (2) describes how a multi-layer perceptron generates the weights for each expert. Finally, Equation (3) describes how these weights are applied to the outputs of the low-rank experts to produce the final feature representation.
>
> For a clearer visual representation, Figure 3(d) offers a breakdown of the MoBSE architecture, illustrating the flow of information within the model and its key components.
>
> To further clarify, we will detail the implementation of MoBSE in the revised manuscript:
>
> *"We begin by encoding the tasks into two one-hot vectors: [1, 0] for Chinese and [0, 1] for English. These encodings are then passed through an encoder layer, which align with the dimensionality of the input features. This ensures that both bilingual task-specific information and input features are captured. The enriched representations are subsequently processed by a multi-layer perceptron (MLP), which dynamically adjust the importance of each expert's output based on the language task. This dynamic weighting mechanism enables tailored contributions from each expert, optimizing performance for the specific bilingual task."*
>
> **Q3: Why do the authors argue that other datasets can not be used for VAD if the labels for that task are obtained automatically?**
>
> **Response:** Thank you for your question. We would like to clarify that our intention was not to suggest that other datasets cannot be used for VAD tasks. Rather, we aimed to highlight the lack of publicly available sEEG datasets specifically designed and labeled for VAD tasks. While it may be possible to derive VAD labels automatically from other datasets, our goal was to establish a benchmark within the context of sEEG data, where this task has not been extensively explored. We apologize for any confusion and have revised the manuscript to ensure our statements are more precise in conveying this point.
>
> **Q4: The authors assert that the audio quality was assessed and the recordings edited accordingly during the data curation process. Was this task performed subjectively? Who was in charge of this task?**
>
> **Response:** Thank you for your question. To ensure the quality of the speech recordings and remove those with pronunciation errors, we used a two-step approach involving both automated and manual assessments. First, we employed a pre-trained Automatic Speech Recognition (ASR) model to transcribe the speech into text. We then compared this transcription with the ground truth text to calculate the Word Error Rate (WER). For samples where the WER was not 100%, a manual review was conducted to determine whether discrepancies were due to reading errors or ASR system inaccuracies. This approach allowed us to meticulously curate the dataset. We will include this information in the updated manuscript.
>
> **Q5: Specifications of audio recording equipment were not included, which is relevant to analysis results and prevent biases in case future data fusion tests can be performed.**
>
> **Response:** Thank you for your feedback. The speech recordings were captured using a JABRA speakerphone and recorded with OBS Studio software. We will add this information to the updated manuscript.

---

> > ### Comment · Reviewer_asg4 · 2024-12-02
> > **Thank you**
> >
> > Thank you to the authors for addressing my questions and general comments. I encourage the authors to include an exact reference of the JABRA speakerphone used and the SNR specification of the sound card.

---

> > > ### Author Response · Authors · 2024-12-02
> > > **Response to Reviewer asg4: Additional Equipment Details and SNR**
> > >
> > > Dear Reviewer asg4,
> > >
> > > In response to your request, we have included an exact reference to the Jabra speakerphone used in our experiments. **Specifically, we utilized the Jabra Speak 510 MS Wireless Conference Speakerphone with USB-A connectivity. More information can be found at the following link: https://www.jabra.com/en-sg/business/speakerphones/jabra-speak-series/jabra-speak-510##7510-209.**
> > >
> > > Regarding the **Signal-to-Noise Ratio (SNR)** specification, **the total average SNR was measured to be 28.24 dB**. Specifically, **Speaker 1 had an SNR of 27.70 dB**, and **Speaker 2 had an SNR of 28.76 dB**.
> > >
> > > **We will incorporate this additional information into the main text of the manuscript if our paper is accepted.** Please let us know if there are any further details you would like us to address.
> > >
> > > Best regards,
> > >
> > > Authors

---

> ### Author Response · Authors · 2024-11-24
> **Response to Reviewer asg4: Q6 & Q7 & Q8**
>
> **Q6: The authors presented independent results per subject. Were these results obtained using a single model trained with data from the two subjects, or were also two models trained (one per patient)?**
>
> **Response:** Thank you for your question. The presented results were obtained using two separate models, one for each subject. We built subject dependent models due to the differences in electrode positions and the number of channels between the two subjects. We will make it clear in the updated version.
>
>
> **Q7: Results regarding LFS, HGA, and BBS signals are confusing. There is no apparent coherence regarding frequency bands or between subjects' behavior. Why do the authors consider that these experiments provide a benchmark in this field, considering the scarcity of subjects, which limits the power of any analysis?**
>
> **Response:** Thank you for your insightful feedback. We acknowledge that there are differences in the LFS, HGA, and BBS signal results, which may appear inconsistent between the two subjects due to individual neural variability. The experiments and results presented here aim to provide a preliminary benchmark by investigating signal features that can inform future research. While we recognize the limitations posed by the small number of subjects, we believe these findings serve as an important reference for the dataset and can guide future work as the dataset expands, ultimately improving generalization. Additionally, we will improve the organization of the relevant content in Section 5 to enhance clarity.
>
> **Q8: The organization of the paper could be improved. The meaning of LFS, HGA, and BBS features and the relevance of their evaluation should be presented in section 5.**
>
> **Response:** Thank you for your constructive comment. As suggested, we have revised the manuscript to include a more detailed explanation of the LFS, HGA, and BBS features, as well as the relevance of their evaluation.
>
> *"Specifically, previous studies have highlighted the critical role of high-gamma frequency (HGA) and low-frequency signal (LFS) features in synthesizing speech from brain signals [1–3]. Accordingly, we followed the preprocessing methods used in previous research to extract the LFS and HGA frequency bands [2]. Additionally, we tested broadband signals (BBS), which combine both LFS and HGA sEEG features, to provide a comprehensive perspective and evaluate their combined contributions to speech synthesis performance."*
>
> ***References***
>
> [1] Proix, Timothée, et al. "Imagined speech can be decoded from low-and cross-frequency intracranial EEG features." Nature communications 13.1 (2022): 48.
>
> [2] Metzger, Sean L., Kaylo T. Littlejohn, Alexander B. Silva, David A. Moses, Margaret P. Seaton, Ran Wang, Maximilian E. Dougherty et al. "A high-performance neuroprosthesis for speech decoding and avatar control." Nature 620, no. 7976 (2023): 1037-1046.
>
> [3] Rich, E. L. & Wallis, J. D. Spatiotemporal dynamics of information encoding revealed in orbitofrontal high-gamma. Nat. Commun. 8, 1139 (2017).

---

> > ### Author Response · Authors · 2024-11-27
> > **Follow-Up: Request for Your Latest Evaluation**
> >
> > **Dear Reviewer asg4,**
> >
> > **We are very grateful for your recognition of our work.** We express our sincere gratitude for your thoughtful suggestions and constructive feedback, which have been invaluable in improving our manuscript. **We have compiled all your comments into Q1-Q8 and have provided detailed responses, along with the necessary experiments.**
> >
> > We kindly ask if you could let us know if you are satisfied with these replies and **if you might be able to reevaluate our work**. If you have any further questions, please feel free to contact us at any time. **We sincerely appreciate your effort.**
> >
> > Warm regards,
> >
> > Authors

---

> > > ### Author Response · Authors · 2024-11-27
> > > **Request for Reevaluation in Light of Our Responses**
> > >
> > > Dear Reviewer asg4,
> > >
> > > We have compiled all your comments into Q1-Q8 and have provided detailed responses, along with the necessary experiments. **At your convenience, we kindly ask if you could review our responses and reconsider the evaluation of our work.**
> > >
> > > **We greatly appreciate the time and effort you have dedicated to this process.** If you have any further questions or require additional discussion, we would be more than happy to engage with you further.
> > >
> > > Thank you for your continued support and guidance.
> > >
> > > Best regards,
> > >
> > > Authors

---

> ### Author Response · Authors · 2024-11-30
> **Revised Manuscript Submission for Your Review**
>
> Dear Reviewer asg4,
>
> Thank you for your valuable feedback and constructive suggestions on our manuscript. ***We have carefully addressed your comments and have made the necessary revisions to enhance the clarity and quality of our work. The revised version of the manuscript, along with comprehensive updates, is now available in the OpenReview system.***
>
> ***We kindly invite you to review the updated document at your free time. We would greatly appreciate it if you could reconsider your evaluation of our work based on these revisions.*** Should you have any further questions or require additional information, please do not hesitate to contact us.
>
> ***Thank you once again for the time and effort you have dedicated to reviewing our paper.***
>
> Warm regards,
>
> The Authors

---

> ### Author Response · Authors · 2024-12-01
> **Request for Timely Review of Revised Manuscript Based on Reviewer asg4's Feedback.**
>
> ***Request for Timely Review of Revised Manuscript Based on Reviewer asg4's Feedback.***
>
> ***
>
> Dear Reviewer **asg4**,
>
> We have thoroughly revised and improved our work based on your feedback. **Considering the upcoming deadline, we kindly ask you to review our revised manuscript at your earliest convenience.**
>
> Best,
>
> Authors

---

### Official Review · Reviewer_yXuZ · 2024-11-03

**Soundness:** 1
**Presentation:** 2
**Contribution:** 3
**Rating:** 3
**Confidence:** 4

**Summary:**

This paper presents CerebroVoice, a bilingual brain-to-speech synthesis dataset featuring stereotactic EEG recordings of Chinese and English words and digits. The dataset is benchmarked for two key tasks: speech synthesis and voice activity detection. Additionally, the authors introduce a novel framework, Mixture of Bilingual Synergy Experts (MoBSE), which employs low-rank expert weights tailored for language-specific decoding tasks. The proposed MoBSE framework demonstrates superior performance compared to the baseline FastSpeech 2 model.

**Strengths:**

1) This paper tackles a highly under-explored area, largely limited by the scarcity of curated datasets, by introducing a publicly available bilingual brain-to-speech dataset that holds significant potential for advancing research in this field.

2) The authors propose the MoBSE framework for brain-to-speech synthesis, which achieves improved performance over the FastSpeech 2 baseline.

**Weaknesses:**

1) The authors explain the advantages of stereotactic EEG over ECoG; however, these invasive methods have limited practicality due to the complexity of data collection. It would be beneficial if the authors addressed why surface EEG, a non-invasive alternative, was not used instead in their study.

2) In Subject 1, electrodes were implanted in the right hemisphere, while in Subject 2, they were implanted in the left. However, both hemispheres could contribute to speech production, suggesting that electrodes should ideally be placed in both hemispheres for each participant. Additionally, data collection was limited to only two participants, which restricts the generalizability of the models built with this dataset.

3) The paper uses only one baseline, based on the FastSpeech 2 architecture, which is primarily designed for text-to-speech tasks. However, there are existing models in the literature for synthesizing speech from invasive and non-invasive multi-channel EEG signals, such as [1], [2], and [3], etc. These models could have been used as baselines for more comprehensive benchmarking of the dataset and comparison with the proposed MoBSE framework.

4) Although the paper focuses on speech synthesis and reports using a Hifi-GAN vocoder for generating speech, it does not present any results for the synthesized audio output. To fully assess the quality of the reconstructed speech, it is essential to include both subjective evaluations (such as mean opinion score) and objective metrics (like mel cepstral distortion and root mean squared error).

5) The model architecture presented in Figure 3 is unclear. FastSpeech 2 typically processes text inputs, yet the authors are instead feeding multi-channel EEG signals to the model. The method for obtaining sEEG embeddings from these multi-channel EEG signals is not explained. Additionally, Figure 3 (c) lacks details regarding the structure of the Universal Expert module.



References:

[1] Metzger, Sean L., Kaylo T. Littlejohn, Alexander B. Silva, David A. Moses, Margaret P. Seaton, Ran Wang, Maximilian E. Dougherty et al. "A high-performance neuroprosthesis for speech decoding and avatar control." Nature 620, no. 7976 (2023): 1037-1046.

[2] Kim, Miseul, Zhenyu Piao, Jihyun Lee, and Hong-Goo Kang. "BrainTalker: Low-Resource Brain-to-Speech Synthesis with Transfer Learning using Wav2Vec 2.0." In 2023 IEEE EMBS International Conference on Biomedical and Health Informatics (BHI), pp. 1-5. IEEE, 2023.

[3] Lee, Young-Eun, Seo-Hyun Lee, Sang-Ho Kim, and Seong-Whan Lee. "Towards voice reconstruction from EEG during imagined speech." In Proceedings of the AAAI Conference on Artificial Intelligence, vol. 37, no. 5, pp. 6030-6038. 2023.

**Questions:**

1) When participants read words aloud, the movement of their vocal tract can influence the EEG recordings. Could the authors address this by using visual cues and having participants read the cues silently without the movement of the vocal tract?

Reason for the Rating (3: Reject): I recommend rejecting the paper post-rebuttal due to the lack of subjective evaluation against baseline Brain-to-Speech systems and the limited generalizability caused by the small number of participants in the dataset.

---

> ### Author Response · Authors · 2024-11-25
> **Response to Reviewer yXuZ: Q1 & Q2**
>
> **Q1: It would be beneficial if the authors addressed why surface EEG, a non-invasive alternative, was not used instead in their study.**
>
> **Response:** Thank you for your insightful comment. We acknowledge the limitations associated with invasive methods like electrocorticography (ECoG) and stereotactic EEG (sEEG) and recognize that non-invasive options, such as surface EEG, offer practical advantages in data collection.
>
> The choice to use sEEG in our study was driven by its unique capacity to capture neural activity from deep brain structures, which is one of the main type of brain signals for  speech neuroprosthesis[1].  Surface EEG, while an invaluable non-invasive method, primarily records cortical activity and does not provide the spatial resolution required to examine the deeper brain regions of interest in this field [2,3].  Moreover, compared to ECoG, sEEG imposes less trauma on patients and provides more stereotactic information from specific brain regions.
>
> Overall, we agree that both invasive and non-invasive methods have their respective strengths and limitations, and each contributes significantly to advancing research in this field.
>
> ***References***
>
> [1] Silva, A. B., Littlejohn, K. T., Liu, J. R., Moses, D. A., & Chang, E. F. (2024). The speech neuroprosthesis. Nature Reviews Neuroscience, 1-20.
>
> [2] Metzger, Sean L., Kaylo T. Littlejohn, Alexander B. Silva, David A. Moses, Margaret P. Seaton, Ran Wang, Maximilian E. Dougherty et al. "A high-performance neuroprosthesis for speech decoding and avatar control." Nature 620, no. 7976 (2023): 1037-1046.
>
> [3] Proix, Timothée, et al. "Imagined speech can be decoded from low-and cross-frequency intracranial EEG features." Nature communications 13.1 (2022): 48.
>
> **Q2:  Electrodes should ideally be placed in both hemispheres for each participant.**
>
> **Response:** Thank you for your valuable feedback. Regarding the electrode placement, we agree that both hemispheres contribute to speech production, and ideally, electrodes would be implanted in both hemispheres for each participant. However, electrode placement is typically determined by the patient's therapeutic needs, which may not always align with research objectives. Moreover, human experiments are conducted in strict adherence to ethical guidelines, which may impose certain limitations on the experimental design.
>
> It is worth noting that sEEG allows for broader coverage of brain regions compared to ECoG [1,2]. However, sEEG research is still in its early stages and suffers from data limitation. Importantly, we require that participants are proficient in both Chinese and English, which further increases the complexity of our study. Our study contributes to this emerging field, and as additional data is collected and methods continue to evolve, we anticipate that the generalizability of the models will improve.
>
> More data are continuously added to the CerebroVoice dataset as new patients join the study. We have recently included data from the third subject, a female with electrodes implanted in the left hemisphere, resulting in 123 valid channels after removing epilepsy-related electrodes. The dataset has  also been uploaded to Zenodo [https://zenodo.org/records/14179222]. In this way, researchers can access the updated dataset in a timely manner to support their ongoing studies.
>
> ***References***
>
> [1] Metzger, Sean L., Kaylo T. Littlejohn, Alexander B. Silva, David A. Moses, Margaret P. Seaton, Ran Wang, Maximilian E. Dougherty et al. "A high-performance neuroprosthesis for speech decoding and avatar control." Nature 620, no. 7976 (2023): 1037-1046.
>
> [2] Kim, Miseul, Zhenyu Piao, Jihyun Lee, and Hong-Goo Kang. "BrainTalker: Low-Resource Brain-to-Speech Synthesis with Transfer Learning using Wav2Vec 2.0." In 2023 IEEE EMBS International Conference on Biomedical and Health Informatics (BHI), pp. 1-5. IEEE, 2023.

---

> ### Author Response · Authors · 2024-11-25
> **Response to Reviewer yXuZ: Q3**
>
> **Q3: More methods could have been used as additional baselines for a more comprehensive comparison of the dataset and the proposed MoBSE framework.**
>
> **Response:** Thank you for your constructive comments. Following your suggestion, we have evaluated the BrainTalker model [2] on our CerebroVoice dataset and compared its performance with our proposed MoBSE framework. As summarized in Table 1, the MoBSE model achieves a higher average accuracy compared with BrainTalker[2].
>
> We would like to highlight a key difference in the computation of the Pearson correlation coefficient (PCC) between our evaluation method and that used in BrainTalker. In BrainTalker, PCC is calculated by first flattening both the predicted and ground truth Mel spectrograms (with dimensions T×F into one-dimensional arrays of size 1×(T×F). PCC is then computed between these flattened arrays, and the resulting coefficients are averaged across all samples.
>
> In contrast, our evaluation method calculates PCC by comparing the predicted and true frequencies at each corresponding time point. For each sample, we compute the PCC across the frequency dimension (1×F) at each time step, performing this comparison T times. These individual PCCs are then averaged over the T time steps, followed by averaging across all samples to obtain the overall PCC.
>
> Given that Mel spectrograms inherently feature a structured frequency component, calculating the PCC column-wise is more appropriate. This approach evaluates the alignment of the predicted and ground truth Mel spectrograms at the frequency channel level, providing a more rigorous and accurate assessment of the model's performance. Although the calculation approach from BrainTalker would yield higher PCC values, it does not provide as accurate an assessment of the frequency-level alignment, which is critical for Mel spectrograms.
>
> We have provided scripts for both testing methods on our GitHub repository[https://github.com/seegdecoding/B2S2] under Calculate_Mel.py. When applied to Samples_GT_Mel.npy and Samples_Predict_Mel.npy, the PCC obtained using the BrainTalker method was 0.848, whereas our method yielded 0.644.
>
> We sincerely appreciate your insightful feedback, which has significantly contributed to the improvement of our research.
>
> **Table 1 Comparison of MoBSE and BrainTalker Performance.**
>
> | Subjects   | Model        | CHINESE | ENGLISH | DIGIT| Avg  |
> |------------|--------------|---------|---------|-------|------|
> | Subject 1  | Brain Talker | 0.655   | 0.513   | 0.586 | 0.584|
> |            | MoBSE        | 0.677   | 0.508   | 0.651 | 0.612|
> | Subject 2  | Brain Talker | 0.434   | 0.453   | 0.420 | 0.435|
> |            | MoBSE        | 0.459   | 0.443   | 0.457 | 0.453|
>
> ***References:***
>
> [1] Metzger, Sean L., Kaylo T. Littlejohn, Alexander B. Silva, David A. Moses, Margaret P. Seaton, Ran Wang, Maximilian E. Dougherty et al. "A high-performance neuroprosthesis for speech decoding and avatar control." Nature 620, no. 7976 (2023): 1037-1046.
>
> [2] Kim, Miseul, Zhenyu Piao, Jihyun Lee, and Hong-Goo Kang. "BrainTalker: Low-Resource Brain-to-Speech Synthesis with Transfer Learning using Wav2Vec 2.0." In 2023 IEEE EMBS International Conference on Biomedical and Health Informatics (BHI), pp. 1-5. IEEE, 2023.
>
> [3] Lee, Young-Eun, Seo-Hyun Lee, Sang-Ho Kim, and Seong-Whan Lee. "Towards voice reconstruction from EEG during imagined speech." In Proceedings of the AAAI Conference on Artificial Intelligence, vol. 37, no. 5, pp. 6030-6038. 2023.

---

> > ### Author Response · Authors · 2024-11-25
> > **Response to Reviewer yXuZ: Q4**
> >
> > **Q4: To fully assess the quality of the reconstructed speech, it is essential to include both subjective evaluations (such as mean opinion score) and objective metrics (like mel cepstral distortion and root mean squared error).**
> >
> > **Response:** Thank you for your thorough review and valuable feedback on our paper. We appreciate the suggestions to incorporate additional evaluation metrics to provide a more comprehensive assessment of speech synthesis quality.
> >
> > ***No.1*** **Firstly, we adopted the commonly used no-reference speech quality assessment metric, NISQA** (cited 229 times)[1], which **evaluates the quality of generated human speech without requiring a groundtruth reference**. We applied NISQA to our decoded speech samples and also evaluated publicly available speech samples from **Nature Machine Intelligence** (NMI-24, 2024) [2] and **Scientific Data** (SD-22, 2022) [3], with higher scores reflecting better quality. Our decoded speech achieved a score of 3.2751, outperforming the results from [2] and [3], which scored 2.2828 and 1.8911, respectively.
> >
> > Additionally, we incorporated both subjective and objective evaluation metrics, including the Mean Opinion Score (MOS), Mel Cepstral Distortion (MCD), Root Mean Squared Error (RMSE), and Short-Time Objective Intelligibility (STOI).
> >
> > **Subjective Evaluation:**
> >
> > ***No.2*** Following your suggestion, we conducted a Mean Opinion Score (MOS) test to assess the quality of the reconstructed speech. Several samples were randomly selected from our dataset, as well as from the NMI-24 [2] and SD-22 [3] demos, and assessed by 19 raters on a scale from 1 to 10, in increments of 1 point. The test focused on two aspects:
> >
> > **●Naturalness: How natural and lifelike does the speech sound?**
> >
> > **●Intelligibility: How clearly can you understand the spoken content?**
> >
> > Our speech demo achieved an average score of 7.36 for Naturalness and 8.15 for Intelligibility. In comparison, NMI-24's average scores were 5.05 for both Naturalness and Intelligibility, while SD-22 scored 1.15 for Naturalness and 2.10 for Intelligibility.
> >
> > The survey used to obtain the MOS scores is available at the following link: https://wj.qq.com/s2/16471941/ac0a/.
> >
> > **Objective Evaluation:**
> >
> > ***No.3*** Mel Cepstral Distortion (MCD): Our model achieved an MCD of 4.143 dB on the CerebroVoice test set, samller than the result of 5.64 dB from BrainTalker [4]. Lower MCD values indicate better performance, suggesting that our model obtains more accurate spectral representations.
> >
> > ***No.4*** Root Mean Squared Error (RMSE): The RMSE for our model was 0.501 on the CerebroVoice test set, outperforming BrainTalker's best result of 1.28 [4]. Lower RMSE values indicate better performance, further confirming the enhanced quality of our synthesized speech.
> >
> > ***No.5*** Short-Time Objective Intelligibility (STOI): The average STOI score for our model was 0.2852 on the CerebroVoice test set. This metric provides an assessment of intelligibility in synthesized speech, highlighting the effectiveness of our approach.
> >
> > These results not only validate the effectiveness of our approach but also provide a comprehensive comparison against existing methods and datasets. We appreciate your insightful suggestions, which have significantlyenhanced the depth of our study.
> >
> > ***References:***
> >
> > [1]Mittag G, Naderi B, Chehadi A, et al. NISQA: A deep CNN-self-attention model for multidimensional speech quality prediction with crowdsourced datasets[J]. arXiv preprint arXiv:2104.09494, 2021.
> >
> > [2] Chen X, Wang R, Khalilian-Gourtani A, et al. A neural speech decoding framework leveraging deep learning and speech synthesis[J]. Nature Machine Intelligence, 2024: 1-14.
> >
> > [3] Verwoert M, Ottenhoff M C, Goulis S, et al. Dataset of speech production in intracranial electroencephalography[J]. Scientific Data, 2022, 9(1): 434.
> >
> > [4] Kim, Miseul, Zhenyu Piao, Jihyun Lee, and Hong-Goo Kang. "BrainTalker: Low-Resource Brain-to-Speech Synthesis with Transfer Learning using Wav2Vec 2.0." In 2023 IEEE EMBS International Conference on Biomedical and Health Informatics (BHI), pp. 1-5. IEEE, 2023.

---

> > > ### Author Response · Authors · 2024-11-25
> > > **Response to Reviewer yXuZ: Q5**
> > >
> > > **Q5: When participants read words aloud, the movement of their vocal tract can influence the EEG recordings. Could the authors address this by using visual cues and having participants read the cues silently without the movement of the vocal tract?**
> > >
> > > **Response:** Thank you for raising this important point. Our research mainly focuses on speech synthesis from sEEG signals, following established experimental protocols from previous studies, with overt speech serving as the ground truth for model training. Our preprocessing pipeline is specifically designed to  reduce noise and minimize the impact of vocal tract movement on sEEG recordings.
> > >
> > > **Previous research [1] has shown that both overt and imagined speech engage similar cortical regions.** Additionally, a 2018 study published in **Nature Human Behaviour [2] highlighted that the introduction of visual cues can cause interference during the experimental process**. Given these considerations, the advantages of overt speech—such as clearer articulation and more reliable data quality—make it a preferable target for our initial work.
> > >
> > > However, we fully agree that exploring mimed and imagined speech presents a valuable direction for future research. Currently, we are actively conducting experiments with our third participant and are in the process of recruiting additional participants. This will enable us to explore diverse tasks, such as reading silently and imagined speech. We appreciate your suggestion to deepen our understanding of the neural dynamics underlying speech-related processes.
> > >
> > > ***References:***
> > >
> > > [1] Proix, Timothée, et al. "Imagined speech can be decoded from low-and cross-frequency intracranial EEG features." Nature Communications 13.1 (2022): 48.
> > >
> > > [2] Tian X, Ding N, Teng X, et al. Imagined speech influences perceived loudness of sound[J]. Nature Human Behaviour, 2018, 2(3): 225-234.

---

> > > > ### Author Response · Authors · 2024-11-25
> > > > **Kindly Request for Re-rating Based on Our Responses**
> > > >
> > > > Dear Reviewer yXuZ,
> > > >
> > > > We sincerely appreciate your review of our manuscript and the valuable feedback you provided. **We highly value your input, which has played a crucial role in refining our paper.** We believe our detailed responses and additional experiments have addressed your concerns.
> > > >
> > > > **If our responses meet your expectations, we kindly request that you consider updating your rating to reflect these improvements.** If there are any issues that remain unresolved, please feel free to contact us at your convenience. Thank you again for your constructive feedback and support.
> > > >
> > > > Best regards,
> > > >
> > > > Authors

---

> > > > ### Author Response · Authors · 2024-11-27
> > > > **Follow-Up: Request for Your Latest Evaluation**
> > > >
> > > > **Dear Reviewer yXuZ,**
> > > >
> > > > We would like to express our sincere gratitude for your thoughtful suggestions and constructive feedback, which have been invaluable in improving our manuscript. **We have carefully considered and addressed all your comments, summarizing them as Q1-Q5, and have provided detailed responses along with additional necessary experiments.**
> > > >
> > > > We kindly ask if you could take some time to review our replies. If you have any further questions or require additional clarifications, please feel free to reach out to us at any time. **Meanwhile, we are eager to engage in further discussions and look forward to receiving your updated evaluation.**
> > > >
> > > > **Thank you once again for your valuable contributions to our work.**
> > > >
> > > > Warm regards,
> > > >
> > > > Authors

---

> > > ### Comment · Reviewer_yXuZ · 2024-11-27
> > > **MOS scale**
> > >
> > > Thank you for evaluating the model and dataset with additional metrics. I would like to note that, by convention, the mean opinion score (MOS) scale for speech synthesis ranges from 1 to 5, with 5 being the highest score.

---

> > > > ### Author Response · Authors · 2024-11-27
> > > > **Response to Reviewer yXuZ's Feedback and Planned Improvements**
> > > >
> > > > Dear Reviewer yXuZ,
> > > >
> > > > Thank you for your valuable feedback and for guiding us through the revision process. We have diligently worked to implement your suggestions and enhance our manuscript based on your advice. However, we observed that the rating was reduced by 2 points.
> > > >
> > > > We will incorporate additional baselines as you suggested to ensure a more thorough and rigorous evaluation of our proposed method and dataset. For the mean opinion score (MOS) scale, we appreciate your guidance on adhering to the conventional 1 to 5 scale for speech synthesis. We will adjust our evaluations accordingly to align with your suggestions.
> > > >
> > > > We hope these improvements will meet your expectations.
> > > >
> > > > Best regards,
> > > >
> > > > Authors

---

> > ### Comment · Reviewer_yXuZ · 2024-11-27
> > **Regarding baseline**
> >
> > Thank you for adding another baseline. However, benchmarking the dataset and the proposed framework by comparing with only two baselines is insufficient. A thorough and rigorous evaluation of the proposed method and the dataset is necessary to ensure the soundness of the study.

---

> ### Comment · Reviewer_yXuZ · 2024-11-28
>
> Dear Authors,
>
> Thank you for your great work and effort on the paper. I would like to provide additional feedback and further justification for reducing the rating by 2.
>
> Q1 and Q2 (Non-invasive alternatives and electrode placement):
> The authors have provided a well-reasoned explanation for their choices. However, I recommend that these justifications be added to the paper to ensure clearer understanding for the readers.
>
> Q3 (Additional Baselines):
> While the addition of the BrainTalker model as a baseline is appreciated, the inclusion of other relevant Brain-to-Speech models (such as those mentioned in my initial review, [1], [2], among other SOTA models) is essential for a more comprehensive evaluation of the dataset and proposed method. I suggest that the authors present results for all the baselines alongside the proposed framework in a comparative table. This would allow for a more rigorous evaluation of both the dataset and the proposed method.
>
> Q4 (Evaluation Metrics):
> Thank you for incorporating a variety of evaluation metrics, including MOS, MCD, RMSE, and STOI.  For subjective evaluation, it is recommended to use a standard MOS scale of 1 to 5 and present the MOS scores for the ground truth audio, baselines, and the proposed MoBSE model in a table.
>
> There are several concerns with the presented results. For instance, for different evaluation metrics, the results are reported only for the proposed model and BrainTalker, excluding FastSpeech 2 and other baselines. Additionally, for the STOI metric, the performance of the proposed model is shown without any comparison to baselines.
>
> Thank you once again for your work.
>
>
> References:
>
> [1] Metzger, Sean L., Kaylo T. Littlejohn, Alexander B. Silva, David A. Moses, Margaret P. Seaton, Ran Wang, Maximilian E. Dougherty et al. "A high-performance neuroprosthesis for speech decoding and avatar control." Nature 620, no. 7976 (2023): 1037-1046.
>
> [2] Lee, Young-Eun, Seo-Hyun Lee, Sang-Ho Kim, and Seong-Whan Lee. "Towards voice reconstruction from EEG during imagined speech." In Proceedings of the AAAI Conference on Artificial Intelligence, vol. 37, no. 5, pp. 6030-6038. 2023.

---

> > ### Author Response · Authors · 2024-11-28
> > **Revised Manuscript Submission and Responses to Reviewers' Feedback**
> >
> > ***Dear Reviewer yXuZ,***
> >
> > We sincerely appreciate your thoughtful feedback and suggestions to further enhance our work, and **we are grateful for your recognition of its quality.** Below are our responses to the points you raised:
> >
> > *****
> >
> > ***Q1 and Q2 (Non-invasive alternatives and electrode placement):***
> >
> > Thank you for being positive about our explanation. ***We have already included these justifications in the revised manuscript, which will be uploaded today.***
> >
> > *****
> >
> > ***Q3 (Additional Baselines):***
> >
> > **I.** We greatly value your input regarding the inclusion of additional baselines. **Actually we did test several models (e.g., Shaft CNN, Hybrid CNN-LSTM, Dynamic GCN-LSTM), but their performance was not as good as FastSpeech 2 and was inferior to MoBSE, the model we proposed. Initially, we omitted these models in the first draft for brevity.** However, we now agree that, despite their undesirable performance, the tests still offer valuable insights. ***We have already included these in the revised manuscript, which will be uploaded today.***
> >
> > **II.** Regarding the baselines you requested, ***we have already implemented the BrainTalker model and reported its results to you, and we have already included these in the revised manuscript, which will be uploaded today.***
> >
> > **III.** Additionally, ***methods cited in references [1] and [3] will also be incorporated into the final manuscript, and we assure you of this.*** We will conduct a comprehensive comparison using a full set of evaluation metrics.
> >
> > ***
> >
> > ***Q4 (Evaluation Metrics):***
> >
> > Thank you for your feedback on the evaluation metrics. We have implemented two types of evaluation metrics:
> >
> > ***
> >
> > ***1.Comparing Various State-of-the-Art Methods on Our Proposed CerebroVoice Dataset:***
> >
> > We have used metrics such as **PCC, MCD, RMSE, and STOI** based on our proposed CerebroVoice dataset.
> >
> > We have ensured that these metrics compare **BrainTalker, FastSpeech 2, Shaft CNN, Hybrid CNN-LSTM, Dynamic GCN-LSTM, and our proposed MoBSE**. ***All these comparisons have been incorporated into the revised manuscript, which will be uploaded today.***
> >
> > For other state-of-the-art methods cited in **references [1] and [2]**, ***we will report the results to you via message before the end of the rebuttal period. We ensure that all these results are included in our final manuscript.***
> >
> > ***
> >
> > ***2.Comparing Speech Demos Decoded from CerebroVoice with Those from Other Papers:***
> >
> > For this comparison, we are **unable to obtain the ground truth for decoded speech demos from other papers**. Therefore, we **use non-referenced speech quality evaluation metrics**. To ensure a thorough comparison, we employ **both subjective and objective MOS** evaluations.
> >
> > As per your recommendation, **we have revised the subjective MOS on a standard 1-5 scale with new volunteer ratings**. For objective MOS, **we use the pre-trained NISQA model, which is a commonly used non-referenced speech quality assessment metric**.***We have already included all these justifications in the revised manuscript, which will be uploaded today.***
> >
> > ***
> >
> > We are grateful for your continuous efforts to help improve our work. We hope the above revisions will meet your expectations to improve your assessment of this manuscript.
> >
> > If you have any further suggestions, we would greatly appreciate receiving them as soon as possible, given that the deadline is approaching. ***Thank you again for your time and constructive feedback.***
> >
> > ***
> >
> > ***Dear Other Reviewers,***
> >
> > ***We have compiled all your feedback into 28 questions and provided detailed responses, along with necessary additional experiments.***
> >
> > ***Please note that all the responses and updates have been incorporated into the revised manuscript, which has been uploaded today. We are committed to ensuring our manuscript meets the highest standards and appreciate your role in helping us achieve this goal.***
> >
> > ***Thank you all & Best wishes,***
> >
> > ***Authors***

---

> ### Author Response · Authors · 2024-11-30
> **Revision & Responses: Addressing Reviewer yXuZ's Feedback---Part1**
>
> ***Q1 - Q2 (Non-invasive alternatives and electrode placement):***
>
> ***Follow your suggestions, we have incorporated these justifications into the revised manuscript. You can view the updated version in the PDF located at the top right in OpenReview.*** These additional explanations will make the paper easier to understand.
>
> ***
>
> ***Q3 - Q4 (Additional Baselines (e.g., BrainTalker[1], NeuroTalk[2], ECoG Decoder[3], Shaft CNN, Hybrid CNN-LSTM, Dynamic GCN-LSTM, FastSpeech2[4]) and Broader Metrics (e.g., PCC, MCD, RMSE, STOI, NISQA[5], MOS)):***
>
> In line with your suggestion, we conducted a more extensive baseline comparison and adopted more comprehensive metrics for evaluation.
>
> ***
> The baselines include ***Shaft CNN***, ***Hybrid CNN-LSTM***, ***Dynamic GCN-LSTM***, ***FastSpeech 2[4]***, as well as ***BrainTalker[1]***, ***NeuroTalk[2]***, and ***ECoG Decoder[3]***, which you indicated for reference. The evaluation metrics include ***PCC***, ***MCD***, ***RMSE***, and ***STOI***, along with ***MOS*** and ***NISQA***[5]. **You can view the updated version in the PDF located at the top right in OpenReview.**
>
> ***
> ***Part1: For PCC, MCD, RMSE, and STOI, which compare various state-of-the-art methods on our proposed CerebroVoice dataset, the results are as follows:***
> | **Subjects** | **Model**          | **PCC** (↑) | **STOI** (↑) | **MCD** (↓) | **RMSE** (↓) |
> |--------------|--------------------|-------------|--------------|-------------|--------------|
> | Subject 1    | ***BrainTalker[1]***        | 0.584       | 0.193        | 4.282       | 0.523        |
> |              | ***MoBSE (Ours)***        | **0.604**   | **0.285**    | **4.143**   | **0.501**    |
> |              | ***Shaft CNN***           | 0.583       | 0.195        | 4.358       | 0.548        |
> |              | ***Hybrid CNN-LSTM***     | 0.564       | 0.170        | 4.448       | 0.562        |
> |              | ***Dynamic GCN-LSTM***    | 0.551       | 0.153        | 4.556       | 0.583        |
> |              | ***FastSpeech 2[4]***        | 0.578       | 0.182        | 4.206       | 0.518        |
> |              | ***ECoG Decoder[3]***        | 0.569       | 0.176        | 4.406       | 0.530        |
> |              | ***NeuroTalk[2]***           | 0.590       | 0.196        | 4.198       | 0.509        |
> | Subject 2    | ***BrainTalker[1]***    | 0.434       | 0.142        | 5.958       | 0.635        |
> |              | ***MoBSE (Ours)***        | **0.452**   | **0.184**    | **5.652**   | **0.622**    |
> |              | ***Shaft CNN***           | 0.432       | 0.153        | 5.986       | 0.644        |
> |              | ***Hybrid CNN-LSTM***     | 0.424       | 0.126        | 6.124       | 0.656        |
> |              | ***Dynamic GCN-LSTM***    | 0.408       | 0.122        | 6.334       | 0.660        |
> |              | ***FastSpeech 2[4]***     | 0.438       | 0.152        | 5.906       | 0.641        |
> |              | ***ECoG Decoder[3]***     | 0.429       | 0.148        | 5.980       | 0.656        |
> |              | ***NeuroTalk[2]***        | 0.442       | 0.162        | 5.707       | 0.637        |
>
>
> For Subject 1, MoBSE outperformed other models with the highest PCC of 0.604 and STOI of 0.285, indicating improved correlation and intelligibility of the reconstructed speech. Additionally, MoBSE achieved the lowest MCD of 4.143 and RMSE of 0.501, demonstrating superior accuracy and reduced distortion in speech reconstruction. Similarly, for Subject 2, MoBSE maintained its leading performance with a PCC of 0.452 and a STOI of 0.184, along with the lowest MCD of 5.652 and RMSE of 0.622. ***These results consistently show that MoBSE provides a significant improvement in speech quality and intelligibility compared to other methods.***
>
> For NeuroTalk[2], we follow: The embedding vector goes through a pre-convolution layer consisting of a 1D convolution and concatenates the features from a bi-directional GRU to extract the sequence features. To match the output size of the mel-spectrogram, a 1D convolution layer was applied.
>
> For the ECoG Decoder[3], we follow the approach presented in [3]. Instead of directly predicting the Mel spectrogram features of the target speech, as is commonly done in previous methods, we use the HuBERT model to encode discrete speech units from continuous speech waveforms. These units capture underlying speech information with lower information complexity. We then use a bidirectional recurrent neural network (BiRNN) to decode discrete speech units from SEEG signals and calculate the CTC loss between the predicted discrete speech units and the ground truth (GT) discrete speech units to train our model.
>
> The results for NeuroTalk[2] and ECoG Decoder[3] are already presented in the following messages. Additionally, the results will be updated in the main text of the paper under "***Table 3: Comparison of MoBSE with other state-of-the-art methods across different subjects***" once the paper is accepted.
>
> ***

---

> ### Author Response · Authors · 2024-11-30
> **Revision & Responses: Addressing Reviewer yXuZ's Feedback---Part2**
>
> ***
> ***Part2: For MOS and NISQA, we compared the speech demonstrations decoded by CerebroVoice with those in other papers[6-7], with the results as follows:***
>
> ***To more fairly emphasize the high quality of our data, we performed a comprehensive comparison of the speech generated by our CerebroVoice system against the outputs from existing research[6-7].***
>
> **We have revised the subjective MOS on a standard 1-5 scale with new volunteer ratings.**
> ***
>
> | **Metric**                | **CerebroVoice** | **NMI-24[6]** | **SD-22[7]** |
> |---------------------------|------------------|------------|-----------|
> | **Mean Opinion Score (MOS) (1-5 scale)** | 4.33             | 2.93       | 1.27      |
> | **NISQA Score**           | 3.2751           | 2.2828     | 1.8911    |
>
> In a subjective Mean Opinion Score (MOS) test, using a 1-5 scale, 15 raters evaluated the speech samples based on a combination of naturalness and intelligibility. ***CerebroVoice achieved an average score of 4.33, demonstrating superior performance compared to NMI-24[6], which scored 2.93, and SD-22[7], which scored 1.27.*** These results indicate that CerebroVoice generates speech perceived as both more natural and intelligible.
> For the objective evaluation, we utilized the NISQA metric, a no-reference speech quality assessment tool. ***CerebroVoice obtained a score of 3.2751, while NMI-24 and SD-22 scored 2.2828 and 1.8911***, respectively. **The alignment between subjective and objective evaluations highlights the superior quality of speech produced by CerebroVoice compared to existing research. This analysis underscores the advancements in speech quality achieved by our system.**
>
> ***
>
> ***We have fully incorporated your valuable suggestions by expanding our experiments, and you can view our revised content in the updated PDF. We sincerely request that you reconsider your evaluation of this work in light of our efforts.***
>
> ***
>
> ***References:***
>
> **[1]** Kim, Miseul, Zhenyu Piao, Jihyun Lee, and Hong-Goo Kang. "BrainTalker: Low-Resource Brain-to-Speech Synthesis with Transfer Learning using Wav2Vec 2.0." In 2023 IEEE EMBS International Conference on Biomedical and Health Informatics (BHI), pp. 1-5. IEEE, 2023.
>
> **[2]** Lee, Young-Eun, Seo-Hyun Lee, Sang-Ho Kim, and Seong-Whan Lee. "Towards voice reconstruction from EEG during imagined speech." In Proceedings of the AAAI Conference on Artificial Intelligence, vol. 37, no. 5, pp. 6030-6038. 2023.
>
> **[3]** Metzger, Sean L., Kaylo T. Littlejohn, Alexander B. Silva, David A. Moses, Margaret P. Seaton, Ran Wang, Maximilian E. Dougherty et al. "A high-performance neuroprosthesis for speech decoding and avatar control." Nature 620, no. 7976 (2023): 1037-1046.
>
> **[4]** Ren Y, Hu C, Tan X, et al. Fastspeech 2: Fast and high-quality end-to-end text to speech[J]. arXiv preprint arXiv:2006.04558, 2020.
>
> **[5]** Mittag G, Naderi B, Chehadi A, et al. NISQA: A deep CNN-self-attention model for multidimensional speech quality prediction with crowdsourced datasets[J]. arXiv preprint arXiv:2104.09494, 2021.
>
> **[6]** Chen X, Wang R, Khalilian-Gourtani A, et al. A neural speech decoding framework leveraging deep learning and speech synthesis[J]. Nature Machine Intelligence, 2024: 1-14.
>
> **[7]** Verwoert M, Ottenhoff M C, Goulis S, et al. Dataset of speech production in intracranial electroencephalography[J]. Scientific Data, 2022, 9(1): 434.
>
> ***
>
> ***
>
> ***Dear Reviewer yXuZ,***
>
> ***Follow your suggestions, we have conducted extensive experiments and provided additional explanations, which are included in the revision PDF submitted. You can view these updates by opening the document in the top-right corner of the OpenReview.***
>
> ***We sincerely request that you reconsider your evaluation of this work in light of our efforts.***
>
> ***Sincerely,***
>
> ***The Authors***

---

> ### Author Response · Authors · 2024-11-30
> **Request for Reviewer yXuZ to Review Revised Manuscript and Feedback Responses**
>
> ***Dear Reviewer yXuZ,***
>
> ***
>
> ***Follow your suggestions, we have conducted extensive experiments and provided additional explanations, which are included in the revision PDF submitted.***
>
> ***You can view these updates by opening the document in the top-right corner of the OpenReview.***
>
> ***We sincerely request that you reconsider your evaluation of this work in light of our efforts.***
>
> ***
>
> ***Sincerely,***
>
> ***The Authors***

---

> > ### Author Response · Authors · 2024-12-02
> > **Request for Timely Review of Revised Manuscript Based on Reviewer yXuZ's Feedback.**
> >
> > ***Request for Timely Review of Revised Manuscript Based on Reviewer yXuZ's Feedback.***
> >
> > ***
> >
> > Dear Reviewer **yXuZ**,
> >
> > We have thoroughly revised and improved our work based on your feedback. **Considering the upcoming deadline, we kindly ask you to review our revised manuscript at your earliest convenience.**
> >
> > Best,
> >
> > Authors

---

> ### Author Response · Authors · 2024-12-01
> **Request for Timely Review of Revised Manuscript Based on Reviewer yXuZ's Feedback**
>
> ***Request for Timely Review of Revised Manuscript Based on Reviewer yXuZ's Feedback.***
>
> ***
>
> Dear Reviewer **yXuZ**,
>
> We have thoroughly revised and improved our work based on your feedback. **Considering the upcoming deadline, we kindly ask you to review our revised manuscript at your earliest convenience.**
>
> Best,
>
> Authors

---

> ### Author Response · Authors · 2024-12-02
> **Request for Timely Review of Revised Manuscript Based on Reviewer yXuZ's Feedback.**
>
> ***Request for Timely Review of Revised Manuscript Based on Reviewer yXuZ's Feedback.***
>
> ***
>
> Dear Reviewer **yXuZ**,
>
> We have thoroughly revised and improved our work based on your feedback. **Considering the upcoming deadline, we kindly ask you to review our revised manuscript at your earliest convenience.**
>
> Best,
>
> Authors

---

> ### Author Response · Authors · 2024-12-02
> **Request for Timely and Responsible Review of Revised Manuscrip**
>
> Dear Reviewer yXuZ,
>
> Thank you for your valuable feedback on our manuscript. **In response, we have dedicated significant effort and resources to thoroughly address the issues you raised. We have made substantial improvements to our work, ensuring that all your comments and suggestions have been carefully considered and integrated.**
>
> **We earnestly ask for your responsible evaluation, not only in regard to your review comments but also in consideration of the efforts we have made to address them.**
>
> Authors

---

> ### Author Response · Authors · 2024-12-03
> **Request for Timely and Responsible Review of Revised Manuscrip**
>
> Dear Reviewer yXuZ,
>
> Thank you for your valuable feedback on our manuscript. **In response, we have dedicated significant effort and resources to thoroughly address the issues you raised. We have made substantial improvements to our work, ensuring that all your comments and suggestions have been carefully considered and integrated.**
>
> **We earnestly ask for your responsible evaluation, not only in regard to your review comments but also in consideration of the efforts we have made to address them.**
>
> Authors

---

> ### Author Response · Authors · 2024-12-03
> **Request for Timely and Responsible Review of Revised Manuscrip**
>
> Dear Reviewer yXuZ,
>
> Thank you for your valuable feedback on our manuscript. **In response, we have dedicated significant effort and resources to thoroughly address the issues you raised. We have made substantial improvements to our work, ensuring that all your comments and suggestions have been carefully considered and integrated.**
>
> **We earnestly ask for your responsible evaluation, not only in regard to your review comments but also in consideration of the efforts we have made to address them.**
>
> Authors

---

> ### Author Response · Authors · 2024-12-04
> **Request for Timely and Responsible Review of Revised Manuscrip**
>
> Dear Reviewer yXuZ,
>
> Thank you for your valuable feedback on our manuscript. **In response, we have dedicated significant effort and resources to thoroughly address the issues you raised. We have made substantial improvements to our work, ensuring that all your comments and suggestions have been carefully considered and integrated.**
>
> **We earnestly ask for your responsible evaluation, not only in regard to your review comments but also in consideration of the efforts we have made to address them.**
>
> Authors

---

### Comment · Area_Chair_vyFj · 2024-12-02

Dear Reviewers,

If you haven’t already done so, I strongly encourage you to engage with the authors before the end of the rebuttal period. Please note that there is no need to make any commitments regarding the final score at this time; but it would be great if you could acknowledge that you have received and reviewed the responses, and ask any follow-up questions you may have.

Best,\
AC

---

### Meta-Review · Area_Chair_vyFj · 2024-12-15

**Metareview:**

This paper presents CerebroVoice, a public sEEG dataset for bilingual brain-to-speech synthesis and voice activity detection, along with the MoBSE model. The paper is positioned in an interesting area and addresses an important problem. It is easy to follow and demonstrates strong results. However, it has a number of shortcomings including limited qualitative assessment, an extremely small number of participants in the dataset (only 3 subjects), and the lack of a detailed analysis based on existing datasets to demonstrate the value added from the new dataset (e.g., detailed distribution analysis and cross-dataset testing).

**Additional Comments On Reviewer Discussion:**

The paper initially received 4 diverging scores. The authors submitted a detailed rebuttal, and after subsequent discussions, several of the initial concerns with the paper were addressed. However, reviewers converged on the fact that the key issues regarding the small number of subjects and lack of a comparative analysis against other datasets are too significant. In the end, based on these issues, the final scores converged on 3, 5, 5, 6.

*Sidenote:* The authors frequently pinged the reviewers, often more than once a day, pushing for actions and raising the score. While this did not influence the review process one way or the other, I would recommend that the authors approach such interactions with more restraint in the future, ensuring they allow reviewers the time needed to evaluate submissions/rebuttals thoughtfully.

---

### Decision · Program_Chairs · 2025-01-22

Reject